# *Leishmania* sand fly-transmission is disrupted by *Delftia tsuruhatensis* TC1 bacteria

Pedro Cecilio [1] ✉, Luana A. Rogerio[2], Tiago D. Serafim [2], Kristina Tang[2], Laura Willen [2], Eva Iniguez[2], Claudio Meneses[2], Luis F. Chaves [3], Yue Zhang[4], Luiza dos Santos Felix[2], Wei Huang [5], Melina Garcia Guizzo[1], Pablo Castañeda-Casado[6], Marcelo Jacobs-Lorena[7], Jesus G. Valenzuela [2], Janneth Rodrigues [6] ✉ & Fabiano Oliveira [2,8] ✉

Most human pathogenic *Leishmania* species are zoonotic agents; therefore, sand fly-based control strategies are essential to prevent parasite circulation. Here, we used the *Delftia tsuruhatensis* TC1 strain, that inhibits the development of *Plasmodium* in mosquitoes, but in the context of *Leishmania*-infected sand flies. We show that *D. tsuruhatensis* TC1 colonizes the midgut of *Phlebotomus duboscqi* sand flies and impacts the development of *L. major* parasites, independently of the colonization timing. This phenotype is likely an indirect consequence of *D. tsuruhatensis* colonization, related with the induction of sand fly gut dysbiosis. Importantly, *Leishmania*-infected, *D. tsuruhatensis*-fed sand flies are less able to transmit *L. major* parasites and cause disease in mice. Modelling supports the disruption of disease endemicity in the field, highlighting *D. tsuruhatensis* as a promising agent for the control of leishmaniasis.

Leishmaniasis is a spectrum of diseases caused by around 20 *Leishmania* species, transmitted via sand fly bites[1]. All but one of these parasite species are zoonotic agents[1,2], highlighting the need for a control strategy not solely focused on the treatment of active cases. Indeed, vector-based interventions are essential to break parasite circulation cycles, toward disease control/elimination. However, the vector control approaches available today, although relevant, are limited. They are mostly based on insecticides - whose effects are restricted to non-resistant sand flies, in confined spaces - and thus, aim to decrease the probability of contact between sand flies and humans[3,4]. There are still no vector control strategies available aiming to impact vector competence.

Recently, we identified and characterized a *Delftia tsuruhatensis* strain that promotes the refractoriness of *Anopheles* spp. mosquitoes to *Plasmodium* spp. infection[5]. The phenotype was attributed to the specific targeting of *Plasmodium* ookinetes by harmane, a beta-carboline alkaloid derivative produced by *Delftia tsuruhatensis* TC1[5]. Interestingly, beta-carboline alkaloids have shown some anti-*Leishmania* activity[6]. This, coupled with the fact that, contrary to the development of *Plamodium* parasites in mosquitoes[7], the development of *Leishmania* parasites is restricted to the sand fly midgut[1] (being thus in constant contact with this bacterial strain and its byproducts), made us hypothesize that *D. tsuruathensis* TC1 would also

[1]Vector Biology Section, Laboratory of Malaria and Vector Research, National Institute of Allergy and Infectious Diseases, National Institutes of Health, Rockville, MD, USA. [2]Vector Molecular Biology Section, Laboratory of Malaria and Vector Research, National; Institute of Allergy and Infectious Diseases, National Institutes of Health, Rockville, MD, USA. [3]Department of Environmental and Occupational Health, School of Public Health-Bloomington, and Department of Geography, Indiana University, Bloomington, IN, USA. [4]Integrated Data Sciences Section (IDSS), Research Technologies Branch, National Institute of Allergy and Infectious Diseases, National Institutes of Health, Rockville, MD, USA. [5]Shanghai Institute of Immunity and Infection, Chinese Academy of Sciences, Shanghai, China. [6]Global Health Medicines R&D, GSK; Tres Cantos, Madrid, Spain. [7]Department of Molecular Microbiology and Immunology, Malaria Research Institute, Johns Hopkins Bloomberg School of Public Health, Baltimore, MD, USA. [8]International Center of Excellence in Research, National Institute of Allergy and Infectious Diseases, Phnom Penh, Cambodia. ✉e-mail: pedro.amadocecilio@nih.gov; janneth-fatima-indira.x.rodrigues@gsk.com; loliveira@niaid.nih.gov

be a good vector refractoriness-promoting agent for the control of leishmaniasis. In this exploratory study, addressing this hypothesis, we investigated whether *D. tsuruathensis* TC1 can be used to impact the transmission of *Leishmania* parasites by sand flies.

## Results

### *Delftia tsuruhatensis* sand fly-midgut-colonization and impact on *Leishmania* infection

We first investigated whether *D. tsuruhatensis* TC1 could colonize the midgut of *Phlebotomus duboscqi* sand flies, the main West-African vector of cutaneous leishmaniasis (CL)[8]. Sand flies were allowed to ingest GFP-*D. tsuruhatensis* (via sugar meal), followed or not by a bloodmeal 24 h later. GFP-signal was detected in the midguts of bacteria-fed sand flies, and not in those of the sugar-fed-only control counterparts (Fig. 1a, b). Even in the absence of a bloodmeal, GFP-signal was detected up to 12-days post-bacterial feeding, suggesting a stable establishment of *D. tsuruhatensis* in the sand fly midgut (Fig. 1a). Moreover, as observed in mosquitoes[5], GFP-*D. tsuruhatensis* expanded in the presence of blood, leading to sand fly midguts with visually higher fluorescence intensity 7-, and 11-days post-bloodmeal (8-, and 12-days post-bacterial feeding; Fig. 1b). Overall, these results suggest that *D. tsuruhatensis* colonizes the sand fly midgut.

Next, we investigated whether *D. tsuruhatensis* colonization was detrimental to the development of *Leishmania* parasites within the sand fly midgut. *P. duboscqi* sand flies were allowed to feed overnight with *D. tsuruhatensis* (wild type TC1; used here and hereafter) via sugar meal, or with sugar only (controls), and infected 24 h later with *L. major*

promastigotes (Fig. 2a). Bacteria-fed sand flies showed a reduction in the total number of parasites per midgut, 7-, and 11-days post-infection; of note, this difference was statistically significant (*versus* controls; p ≤ 0.0005; Fig. 2b), and a dose-dependence tendency was observed in this context (fig. S1). Consequently, the median number of infectious metacyclic parasites was 74%, and 82% lower (respectively) in the midguts of *D. tsuruhatensis*-fed *versus* control sand flies (*p* ≤ 0.0023; Fig. 2c). However, the proportion of metacyclic parasites per total number of parasites was similar in both groups (Fig. 2d), suggesting that a particular parasite stage is not specifically being affected by *D. tsuruhatensis*. Interestingly, when we compared the diameter of the anterior midgut of *D. tsuruhatensis*-fed *versus* control sand flies, we observed that the former showed smaller diameters; such a difference was statistically significant (*p* = 0.0266; Fig. 2e). Overall, these results suggest that the colonization of the sand fly midgut with *D. tsuruhatensis* negatively impacts the development of *Leishmania* parasites within this niche.

### Specificity of the *Delftia tsuruhatensis* effect on *Leishmania* infection

To understand whether our phenotype was *D. tsuruhatensis*-specific, *P. duboscqi* sand flies were allowed to feed on *E. coli* [a component of the gut microbiota of different sand fly species[9]], and infected with *L. major* parasites one day later. The parasite burden of *E. coli*-fed sand flies was similar to that of control sand flies (Fig. 3a), in line with that reported for a different *Leishmania*-sand fly pairing[10]. Additionally, a similar non-significant effect was observed in the context of two

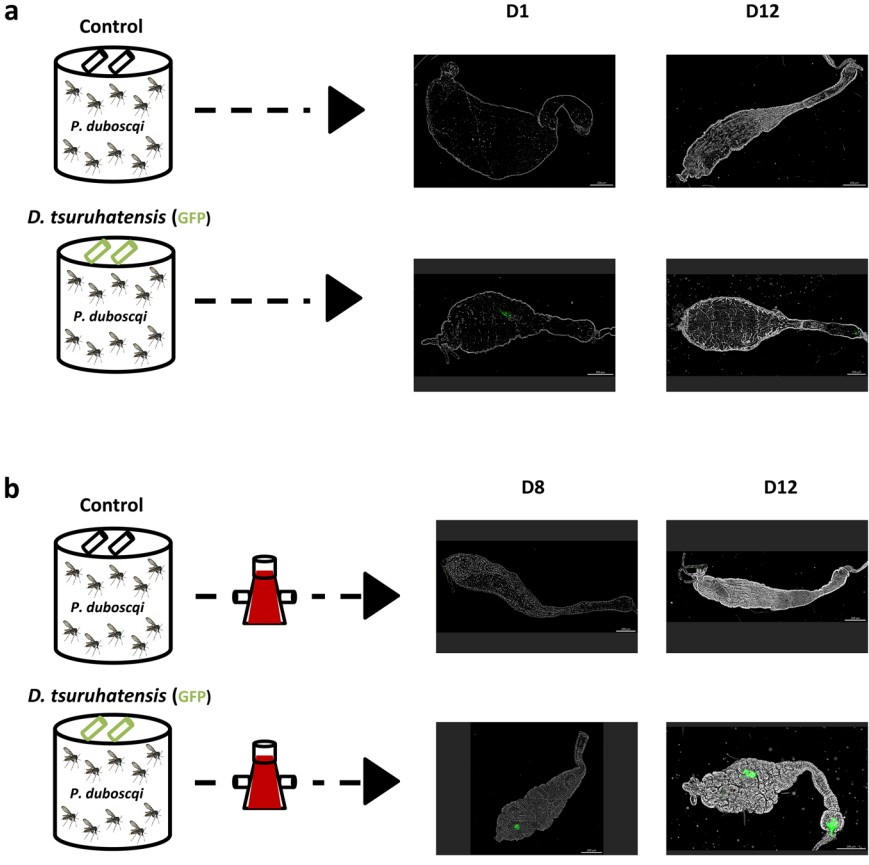

**Fig. 1 | *Delftia tsuruhatensis* colonizes the sand fly midgut. a, b** Sand flies were allowed to take a sugar meal alone (control), or containing GFP-expressing *Delftia tsuruhatensis*, followed or not by a bloodmeal 24 h later. Midguts were dissected at different time-points after bacterial feeding, and GFP (green fluorescent protein) expression was assessed using fluorescence microscopy. **a** Representative images of midguts dissected from sand flies fed only with sugar containing (bottom images) or not (top images) fluorescent bacteria, 1- and 12-days post-feeding. Scale bars=200 μm. **b** Representative images of midguts dissected from sand flies fed first with sugar containing (bottom images) or not bacteria (top images), and then with blood 24 h after, 7, and 11 days post-bloodmeal. Scale bars=200 μm. These results are representative of those obtained in three independent experiments.

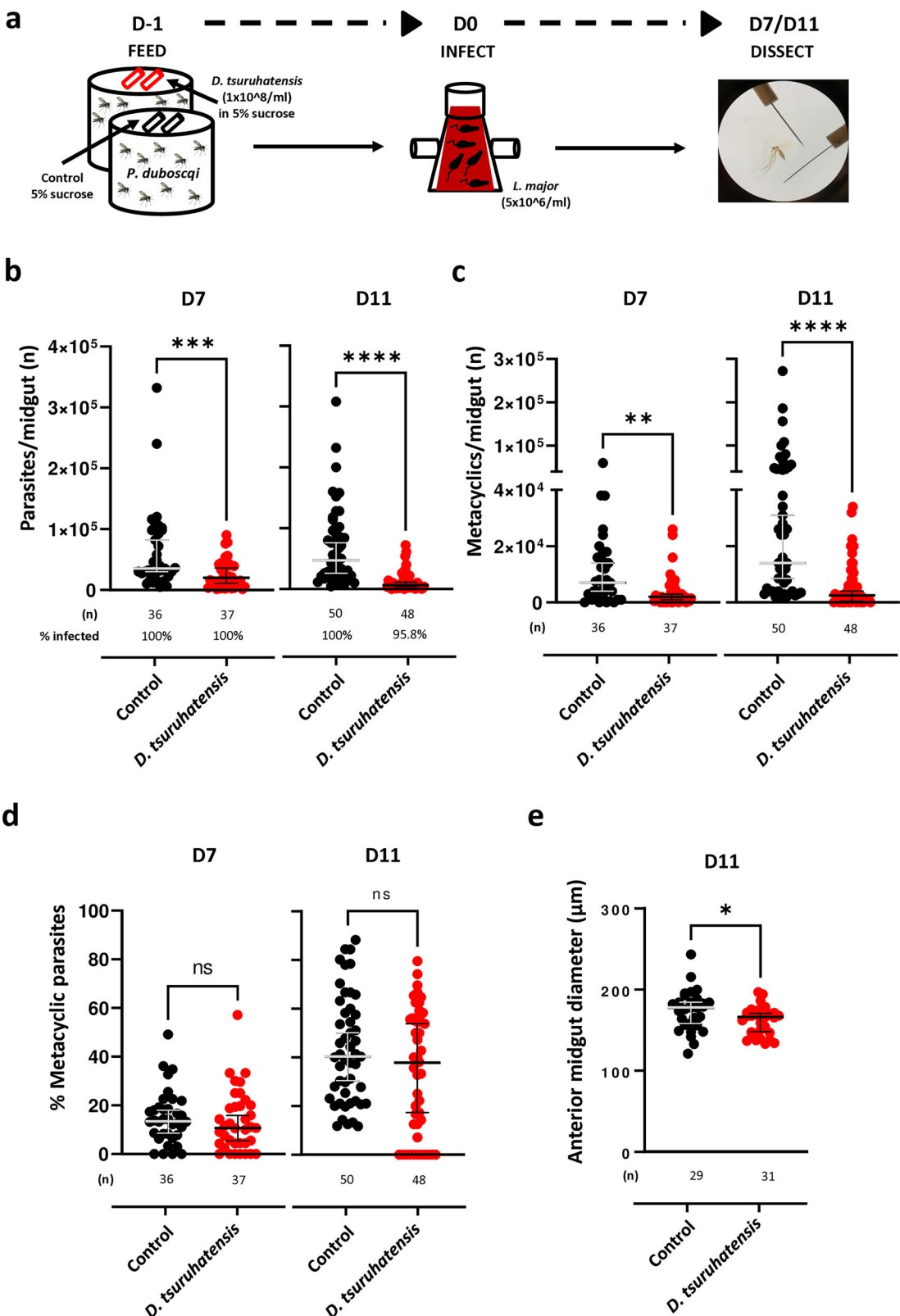

bacterial strains isolated from the gut of *P. duboscqi* sand flies from our colony (identified as *Ornithinbacillus massiliensis* and *Enterococcus faecalis*; Fig. 3b, c). Of note, when we fed the same amount of dead *D. tsuruhatensis* bacteria to *P. duboscqi* sand flies, a significant effect on the development of *Leishmania* parasites was no longer observed (Fig. 3d). Altogether, our data suggest that the impact on parasite development depends on sand fly gut colonization by live *D.*

*tsuruhatensis* bacteria and is probably not a consequence of non-specific bacterial infection and/or of an immune response to bacterial antigens.

Next, we investigated whether the *D. tsuruhatensis* impact on the establishment of *Leishmania* parasites within sand flies was due to bacteria-secreted products. We started by evaluating in vitro the growth of *L. major* parasites cultured with increasing amounts of

**Fig. 2 | *Delftia tsuruhatensis* colonization impacts the establishment of *Leishmania* parasites in the sand fly midgut.** Sand flies were allowed to take a sugar meal (5% sterile sucrose solution), alone (control), or containing *Delftia tsuruhatensis* ($1 \times 10^8$/ml), and then infected with *L. major* parasites via artificial membrane feeding. Midguts were dissected 7 and 11 days post-bloodmeal and the infection burden was quantified microscopically. **a** Schematization of the experimental layout used. **b** Total number of parasites per midgut of control (black dots) and bacteria-fed (red dots) sand flies, 7-, and 11-days post-infection. **c** Total number of metacyclic promastigotes per midgut of control (black dots) and bacteria-fed (red dots) sand flies, 7-, and 11-days post-infection. **d** Percentage of metacyclic promastigotes per midgut of control (black dots) and bacteria-fed (red dots) sand flies, 7-, and 11-days post-infection. **e** Diameter of the anterior midgut of *Leishmania*-infected control (black dots) and bacteria-fed (red dots) sand flies, 11-days post-infection. All results were obtained in, at least, two independent experiments. Each dot represents a single midgut. The group median and 95% CI are also shown, as are the (n), and prevalence of infection, when applicable. Statistical significance was determined using the Mann-Whitney test (**b–d**; two-sided), or Unpaired t-test (**e**; two-sided) and is highlighted: (**b**) ***$p = 0.0005$, and ****$p = 2.334 \times 10^{-12}$; (**c**) **$p = 0.0023$, and ****$p = 8.2315 \times 10^{-9}$; (**c**) ns – not significant ($p > 0.05$); and (**d**) *$p = 0.0266$. Source data are provided as a Source Data file.

bacteria-free or control culture supernatant. Parasites showed impaired growth, when cultured with at least 50% *D. tsuruhatensis*-supernatant (*versus* control supernatant; at least $p \le 0.05$; fig. S2a, b). This effect was due to low-molecular weight components, of less than 10 KDa (Fig. S2c), but was not observed when parasites were cultured with harmane alone (fig. S2d). Of note, harmane and other β-carboline alkaloids have been reported to have some antileishmanial activity in other studies[6], an effect we could not detect in the present study. Therefore, these results suggest that *D. tsuruhatensis* impacts the development of *Plasmodium*[5] and *Leishmania major* parasites distinctly. Of note, oral administration of both harmane (50 μM), and surprisingly, *D. tsuruhatensis*-supernatant (up to 100%) to sand flies did not impact *L. major* parasites in vivo (fig. S2e–g). Overall, our results suggest that live *D. tsuruhatensis* is the best option to consider for the control of *Leishmania* within sand flies.

## Effect of *Delftia tsuruhatensis* colonization on the sand fly gut microbiota

Changes in the sand fly gut microbiota (e.g. after antibiotic treatment) can impact the establishment of *Leismania* parasites in sand flies[11]. Therefore, next we investigated the sand fly gut microbial dynamics. Pools of sand fly midguts were collected one day after *D. tsuruhatensis* (or control sugar) feeding and prior to infection (day 0), as well as 2-, 5-, 7-, 9-, and 12-days after *L. major* infection, and subjected to metagenomics analysis. Strikingly, *D. tsuruhatensis* colonization led to a reshaping of the relative bacterial abundance landscape in the midgut of sand flies throughout the experimental timeline. The gut microbiota of *L. major*-infected control *P. duboscqi* sand flies showed the expected complexity (Fig. 3e), with similarities to that previously reported for *Leishmania infantum*-infected *Lutzomyia longipalpis* sand flies[11]. Conversely, in *D. tsuruhatensis*-fed sand flies a clear prevalence of a single bacterial genus, *Serratia* spp., was evident at all time-points (Fig. 3e). Of note, while the relative abundance of Serratia sp. (identified as Serratia ureilytica via shotgun sequencing; accession number: SAMN40248728) was higher than that of Delftia spp. (fig. S3a), the estimated absolute abundance of both *Serratia* sp. and *Delftia* spp. consistently increased in a similar order of magnitude in bacteria-fed *versus* control sand flies (Fig. 3f). Also relevant, most of the other genera showed a significant lower abundance in *D. tsuruhatensis*-fed *versus* control sand flies, including *Porphyromonas*, *Pantoea*, *Tsukamurella*, and *Ralstonia* (data S1). However, these results did not translate into major changes in alpha-diversity comparing bacteria-fed *versus* control sand flies (fig. S3b–e). Still, when we looked at beta-diversity, a spatially different clustering of *D. tsuruhatensis*-fed and control samples was detected (Fig. 3g), with statistical significance. These results suggest that the gut microbiota composition of bacteria-fed, and control sand flies is different, and consequently that colonization of sand flies with *D. tsuruhatensis* induces gut dysbiosis.

## Impact of *Delftia tsuruhatensis* sand fly colonization on different *Leishmania* artificial-infection contexts

For an intervention with *D. tsuruhatensis* to be successful in the field, it needs to work regardless of the colonization timing. Therefore, next

we investigated whether our phenotype was time-dependent (fig. S4). First, we looked at sand flies infected seven days after bacterial feeding (fig. S4a). We detected a 90% reduction in the number of *Leishmania* parasites per midgut in *D. tsuruhatensis*-fed *versus* control sand flies ($p < 0.0001$; Fig. 4a). Next, we tested whether such a phenotype would be maintained in sand flies colonized after *Leishmania* infection. We infected sand flies with *L. major* parasites and waited 5 days (to allow the defecation of bloodmeal remnants) to introduce *D. tsuruhatensis* via sugar meal (fig. S4b). Seven-days post-infection (two days post-bacterial feeding), the parasite burden in bacteria-fed *versus* control sand flies was similar (Fig. 4b, left graph). However, 4 days later (11 days post-infection/6 days bacterial feeding), a reduction in the number of parasites per midgut in bacteria-fed *versus* control sand flies was observed; this difference was statistically significant (˃ 90% reduction; $p < 0.0001$; Fig. 4b, right graph). These results suggest that *D. tsuruhatensis* impacts the development of *Leishmania* parasites in sand flies, even if they are already infected. Notably, a significant effect was still observed when we fed *D. tsuruhatensis* to sand flies 8 days post-infection (fig. S4c), when infectious metacyclics already populate the sand fly midgut (˃ 50% lower infection burden *versus* controls; $p = 0.0077$; Fig. 4c). Altogether, our results suggest that *Leishmania* parasite reduction in sand flies should occur regardless of the *D. tsuruhatensis* colonization timing.

In nature, sand flies are expected to take multiple blood meals during their life span[1,12]. Importantly, when infected sand flies take subsequent blood meals, *Leishmania* parasites expand within their midgut, positively impacting vector competence[12,13]. Therefore, we evaluated the effect of a second bloodmeal on *D. tsuruhatensis*-colonized, *Leishmania*-infected sand flies. We first introduced *D. tsuruhatensis* (control sand flies received sucrose) and infected the sand flies one day later. We then waited for 11 days for the infection to mature, and finally allowed the sand flies to take a second, non-infected, bloodmeal. Six days after the second bloodmeal, sand flies were dissected (fig. S4d). Strikingly, the phenotype observed was even stronger than that observed previously, with most sand flies in the *D. tsuruhatensis* group showing a very low level of parasites per midgut ($p < 0.0001$; Fig. 4d). Of note, in control sand flies the expected phenotype with a more homogeneous distribution of metacyclic promastigotes[13], was observed. Overall, these results strengthen the claim that *D. tsuruhatensis* will negatively impact the development of *Leishmania* parasites in sand flies in the field.

## Effect of *Delftia tsuruhatensis* colonization on *Leishmania* natural sand fly infection

In nature, sand flies acquire parasites via feeding on infected reservoirs. Therefore, next we used a natural infection model to extrapolate the real impact of *D. tsuruhatensis*. Briefly, we allowed sand flies to feed on active CL lesions in the ears of *L. major*-infected mice and separated the blood fed insects. Half of the blood-fed sand flies were then fed *D. tsuruhatensis* via sugar-meal overnight, while the other half received sucrose solution alone. A significant effect was detected comparing the median infection burden of bacteria-fed *versus* control sand flies ($p = 0.0167$; Fig. 5a). Additionally, the prevalence of infection was 1.6-

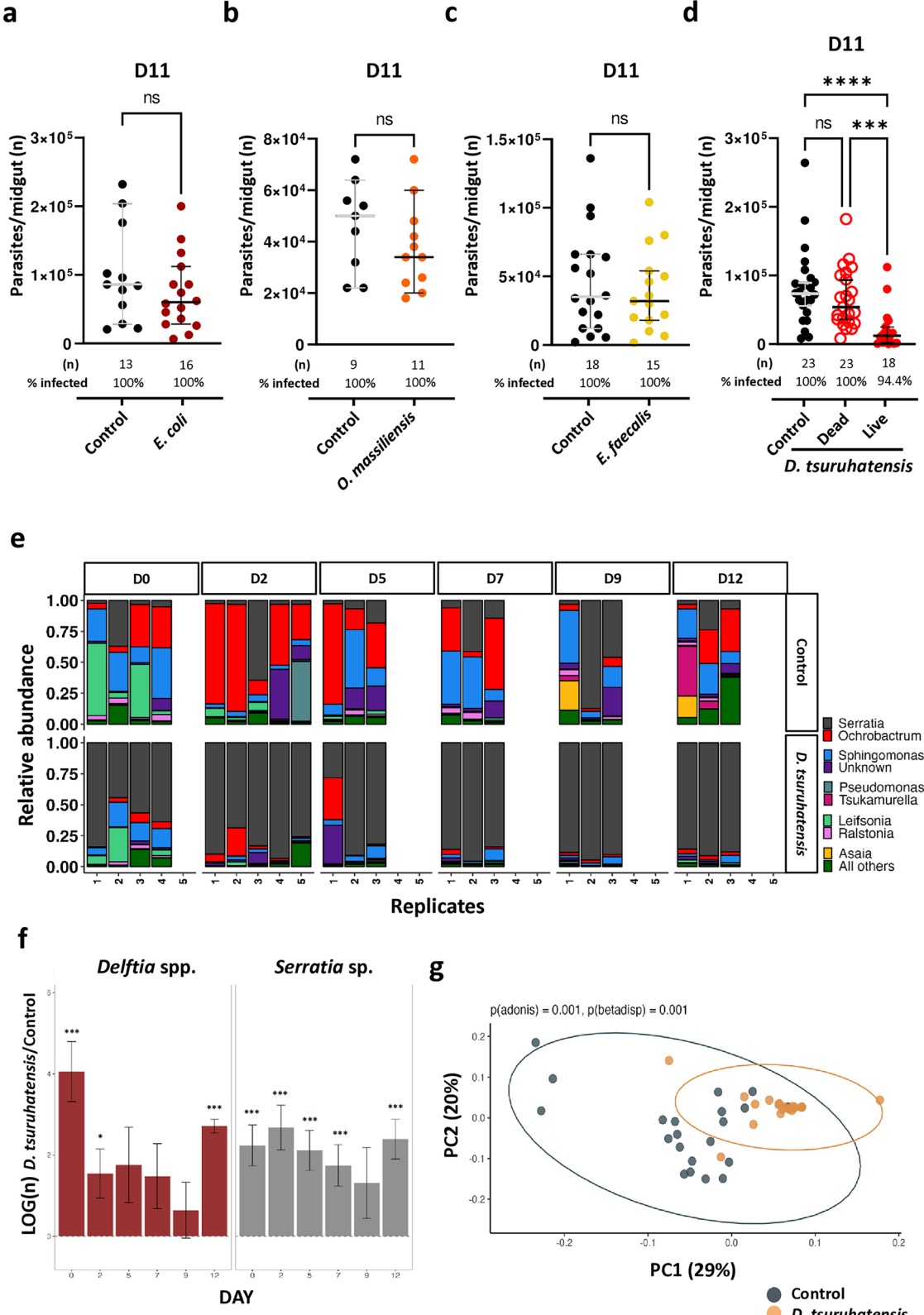

fold lower in bacteria-fed *versus* control sand flies (30.9% *versus* 51.2%; infection relative risk (RR) = 0.6491 - 95% CI of 0.4277 to 0.9820; $p$ = 0.0482; Fig. 5b). This result either suggests that *D. tsuruhatensis* leads to complete parasite elimination in the context of low initial sand fly infection burdens [less than 100 parasites are usually ingested by sand flies in the natural infection context[13]], or that colonization with *D. tsuruhatensis* specifically impacts the fitness of infected sand flies.

## Impact of *Delftia tsuruhatensis* colonization on sand fly mortality

In line with the above, we looked at sand fly survival. Always comparing *D. tsuruhatensis*-fed with control sand flies, we determined the mortality, daily, for sand flies that never took a bloodmeal, as well as for those that took a non-infected or a *Leishmania*-infected bloodmeal. In the absence of blood, mortality was negligible in both *D. tsuruhatensis*-fed and control

**Fig. 3 | The phenotype is *Delftia tsuruhatensis*-specific, and probably secondary to the gut dysbiosis induced by *D. tsuruhatensis* colonization. a–d** Sand flies were allowed to take a sugar meal alone (control) or containing different bacteria and then infected with *L. major* parasites. The midgut infection burden was quantified microscopically at day 11. **a–d** Total number of parasites per midgut of control (black dots) and (**a**) *E. coli*-fed (bordeaux dots), (**b**) *O. massiliensis*-fed (orange dots), (**c**) *E. faecalis*-fed (yellow dots), or (**d**) heat-killed *D. tsuruhatensis*-fed (open red circles), and live *D. tsuruhatensis*-fed (red dots) sand flies. Each dot represents a single midgut; the group median and 95% CI are shown. Statistical significance was determined using the Unpaired t-test (two-sided) (**a–c**) or Kruskal-Wallis test with Dunn's post-hoc analysis (**d**) and is highlighted: (**a–d**) ns – not significant ($p > 0.05$); and (**d**) ***$p = 0.000856$, and ****$p = 5.38433 \times 10^{-5}$. **e–g** Sand flies were allowed to take a sugar meal alone (control), or containing *D. tsuruhatensis*, and then infected with *L. major* parasites. Pools of sand fly midguts were collected one day after bacterial/control feeding and before infection (day 0), as well as 2-to-12 days post-infection and subjected to metagenomics analysis. **e** Relative abundance (genus level) per sample pool/time-point. Control and bacteria-fed samples in top and bottom panels, respectively, with the most abundant genera color-coded. **f** Estimated absolute abundance of *Delftia* spp. (left panel) and *Serratia* spp. (right panel) in *Delftia tsuruhatensis*-treated *versus* control sand flies per time-point. Data are represented by effect size (LogFC) and Standard Error bars (two-sided; Bonferroni adjusted) derived from the ANCOM-BC model. All effect sizes with adjusted $p < 0.05$ are indicated: *, and *** significant at 5%, and 0.1% level of significance, respectively. Exact adjusted p values can be found in Table S2. **g** PCA plot referring to the beta-diversity weighed analysis at the group level (control samples in grey and bacteria-fed samples in gold). Each dot represents and individual sample. Ellipses denote data dispersion at the group level. Statistical significance (permutational ANOVA and beta dispersion analyses) is denoted in the graph. All results were obtained in, at least, two independent experiments. Source data are provided as a Source Data file.

sand flies (Fig. 5c, left panel). In the presence of non-infected blood, mortality increased for both groups (Fig. 5c, middle panel). Interestingly, in the context of infected blood, we saw a clear separation of the survival curves of control and *D. tsuruhatensis*-fed sand flies; the latter showed a lower survival rate throughout the monitoring period (Fig. 5c, right panel). Overall, these data translated into a significant difference in the survival of, only *Leishmania*-infected *D. tsuruhatensis*-fed *versus* control sand flies (data from day 11; $p = 0.0383$; Fig. 5d). Strikingly, the increase in mortality was even more pronounced in infected sand flies that took a second bloodmeal (25% survival *versus* 90% in the control group; mortality RR = 7.216 - 95% CI of 4.224 to 12.78; $p < 0.0001$; Fig. 5e). Therefore, the use of *D. tsuruhatensis* TC1 has the potential, not only to decrease the infection burden of sand flies, but also to reduce the number of infected sand flies in nature.

### Effect of *Delftia tsuruhatensis* on *Leishmania* transmission by sand flies

To understand whether our phenotype was relevant for the transmission of *Leishmania* parasites, we infected *D. tsuruhatensis*-fed and control sand flies in parallel, waited 11 days, and then allowed flies in each group to bite the ears of naïve BALB/c mice (20 sand flies/ear). Expectedly, the percentage of blood-engorged sand flies in the bacteria-fed group was higher than that in the control group; of note, this difference was statistically significant ($p = 0.0005$; Fig. 6a). This aligns with the "blocked fly" hypothesis[14]; the lower infection burden in *D. tsuruhatensis*-colonized sand flies leads to smaller parasite plugs and better feeding success. Importantly, in line with this, the percentage of animals bitten by *D. tsuruhatensis*-fed, *Leishmania*-infected sand flies that developed CL lesions after transmission was much lower than that of animals in the control group (25% *versus* 80%, respectively; disease RR = 0.2361 - 95% CI of 0.08004 to 0.6084; $p = 0.002$; Fig. 6b). Furthermore, lesions in the ears of control animals, were often more widespread (Fig. 6c) and associated with higher parasite burdens ($p < 0.0001$; Fig. 6d). Strikingly, we detected parasites (above the detection limit) in all animals bitten by control sand flies, but only in 26.6% of animals bitten by bacteria-fed sand flies (Fig. 6d). Overall, these results show that, *D. tsuruhatensis* sand fly colonization negatively impacts *Leishmania* transmission and the subsequent development of CL.

### Mathematical modeling to predict the impact of *Delftia tsuruhatensis* in the field

Finally, we used a mathematical model to investigate whether our results could translate into changes in leishmaniasis endemicity in the field. Briefly, our model assessed the effect of a potential intervention, focusing on basic reproduction number ($R_O$) estimates, in the context of three different modelling studies based on field data pertaining to dogs[15,16], multiple host/vector species[17], and humans[18]. Accounting for the increased mortality of *Delftia*-colonized, *Leishmania*-infected sand flies alone, a decrease of $R_O$ below 1 was predicted for 2 of the 3 study populations (Fig. 6e; top panel). Importantly, the decrease of $R_O$ was even more pronounced when we incorporated into the model the observed reduction in the infection burden in *D. tsuruhatensis*-colonized sand flies. An $R_O$ below the threshold of one was predicted for all study populations (Fig. 6e; bottom panel), suggesting that disease endemicity can be disrupted if we promote the colonization of wild sand flies with *D. tsuruhatensis*.

## Discussion

While human interventions should minimize the disruption of ecosystems[19], the targeting of specific disease-transmitting arthropod populations is admissible from a risk-benefit perspective. In fact, many vector control approaches are specifically designed to impact the fitness of disease insect vectors[4,20]. This is particularly true in the context of sand flies. Sand fly-based interventions for the control of leishmaniases rely mostly (if not only) on the use of insecticides, whose efficacy is limited not only because of the emergence of resistance, but also because we still lack basic epidemiologic information[21]. To the best of our knowledge, there are still no strategies that promote sand fly-vector refractoriness for the control of leishmaniasis. Here, we show that *D. tsuruhatensis* TC1 can colonize the gut of sand flies, leading to significant changes that negatively impact the transmission of *Leishmania* to hosts. Importantly, we also show that this bacterial strain seems to impact the fitness of *Leishmania*-infected sand flies alone (accounting to a total of 0.7 to 2% of all sand flies in the field, according to different studies[22]), as opposed to the indiscriminate killing of all sand flies, regardless of the infection status, by insecticides. Therefore, the use of *D. tsuruhatensis* TC1 is a promising vector-based approach for the control of leishmaniasis in the field, with a presumably lower ecological impact than the vector control approaches in effect today.

One curious observation in this study was the dramatic change in the sand fly gut microbiota after *D. tsuruhatensis* colonization. While we were not expecting such a phenotype, our results can be explained. *D. tsuruhatensis* shows a remarkable potential for adaptation[23]; it was reported to inhibit the growth of both environmental fungal/bacterial strains[24], and known human pathogenic bacteria[23,25]. Therefore, we hypothesize that, to colonize the midgut, *D. tsuruhatensis* TC1 outcompetes susceptible bacteria within the sand fly gut microbiota, leading to the expansion of non-susceptible bacteria (e.g., *Serratia ureilytica*). Notably, it was previously reported that sand flies treated with antibiotics lose the ability to support the development of *Leishmania*[11,26]. Therefore, we may assume that the lower *Leishmania* infection burden in bacteria-fed sand flies could be a direct consequence of gut dysbiosis. For instance, the decrease in the abundance of several genera after *D. tsuruhatensis* introduction may translate into the "elimination" of bacteria that promote the growth of *Leishmania* parasites[26]. Conversely, the expansion of *S. ureilytica* after *D. tsuruhatensis* colonization may be detrimental to the development of

*Leishmania* within sand flies (a role of sand fly immunity cannot be excluded in this context). In fact, while there are no data referring to *S. ureilytica* in the context of leishmaniasis, a study showed that *Serratia marcescens* can induce the lysis of *Leishmania* in vitro[27]. *Serratia* spp.

was also reported to impair the development of *Leishmania* parasites in the sand fly midgut[10,28]; still, others showed that *Serratia rubidaea* promotes the growth of *Leishmania* parasites in the context of sand fly gut dysbiosis[26]. Importantly, *Serratia* spp. was reported as part of the

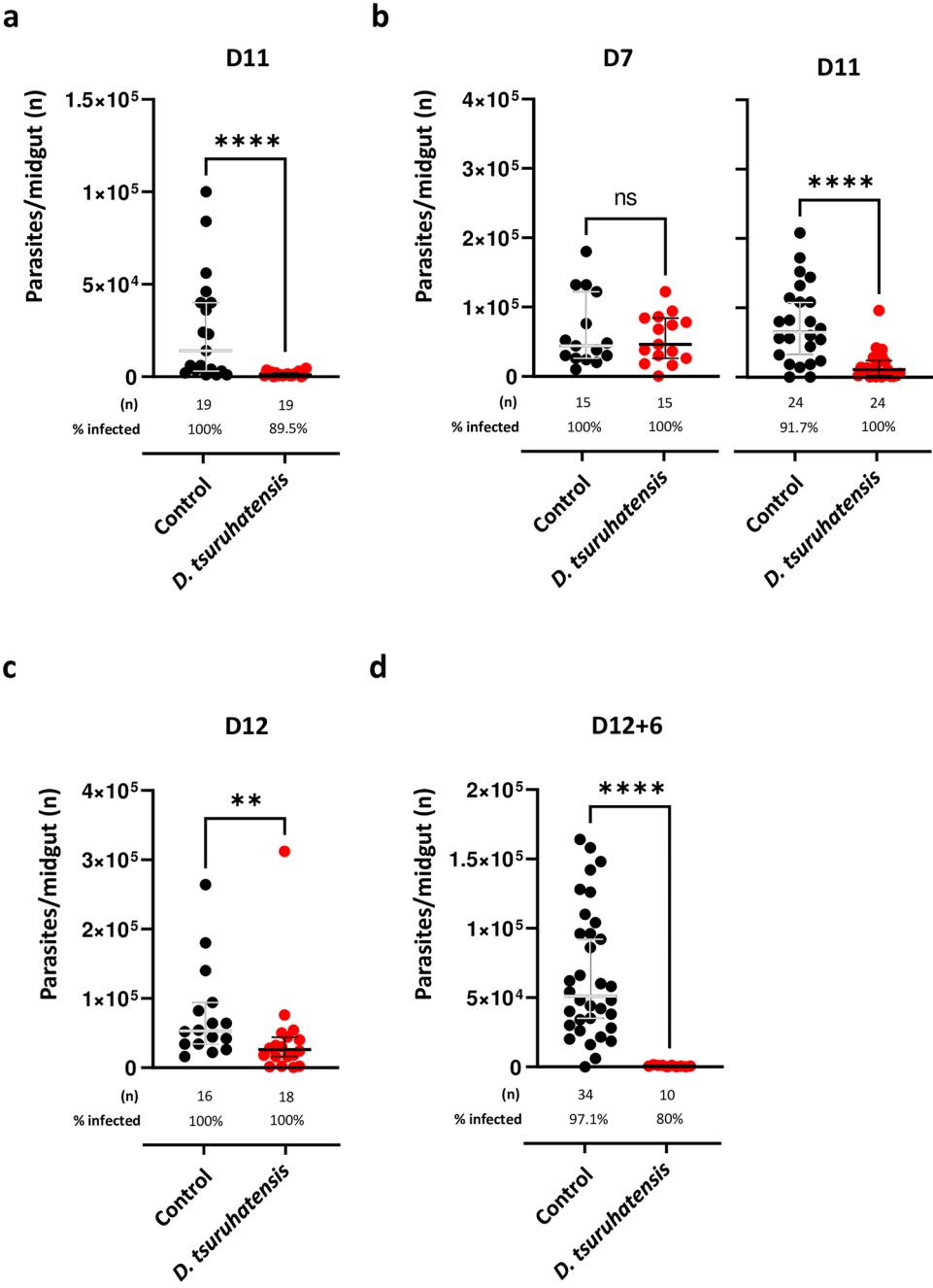

**Fig. 4 | The "anti-*Leishmania*" effect in *Delftia tsuruhatensis*-carrying sand flies is independent of the timing of colonization and retained after a second bloodmeal.** The experimental layouts referring to the data represented here are illustrated in fig. S4. **a** Sand flies were allowed to take a sugar meal alone (control), or containing *D. tsuruhatensis* overnight, given sugar in the next six days, and then infected with *L. major* parasites. The midgut infection burden was quantified microscopically at day 11. The total number of parasites per midgut of control (black dots) and *D. tsuruhatensis*-fed (red dots) sand flies is represented. **b** Sand flies were infected with *L. major* parasites, and then allowed to take a sugar meal, alone (control), or containing *D. tsuruhatensis*, 5 days after infection. The midgut infection burden was quantified microscopically at days 7 and 11. The total number of parasites per midgut of control (black dots) and *D. tsuruhatensis*-fed (red dots) sand flies is represented per time-point. **c** Sand flies were infected with *L. major* parasites, and then allowed to take a sugar meal, alone (control), or containing *D. tsuruhatensis*,

8 days after infection. The midgut infection burden was quantified microscopically at day 12. The total number of parasites per midgut of control (black dots) and *D. tsuruhatensis*-fed (red dots) sand flies is represented. **d** Sand flies were allowed to take a sugar meal, alone (control), or containing *D. tsuruhatensis*, infected with *L. major* parasites via artificial membrane feeding one day later, and given a second, non-infected bloodmeal 12 days after. Midguts were dissected 6 days after the second bloodmeal and the infection burden was quantified microscopically. The total number of parasites per midgut of control (black dots) and *D. tsuruhatensis*-fed (red dots) sand flies is represented. All results were obtained in, at least, two independent experiments. Each dot represents a single midgut. The group median and 95% CI are also shown. Statistical significance was determined using the Mann-Whitney test (two-sided) and is highlighted: (**a**) ****$p = 1.27257 \times 10^{-5}$; (**b**) ns – not significant ($p > 0.05$), ****$p = 2.511 \times 10^{-5}$; (**c**) **$p = 0.0077$; and (**d**) ****$p = 7.29469 \times 10^{-8}$. Source data are provided as a Source Data file.

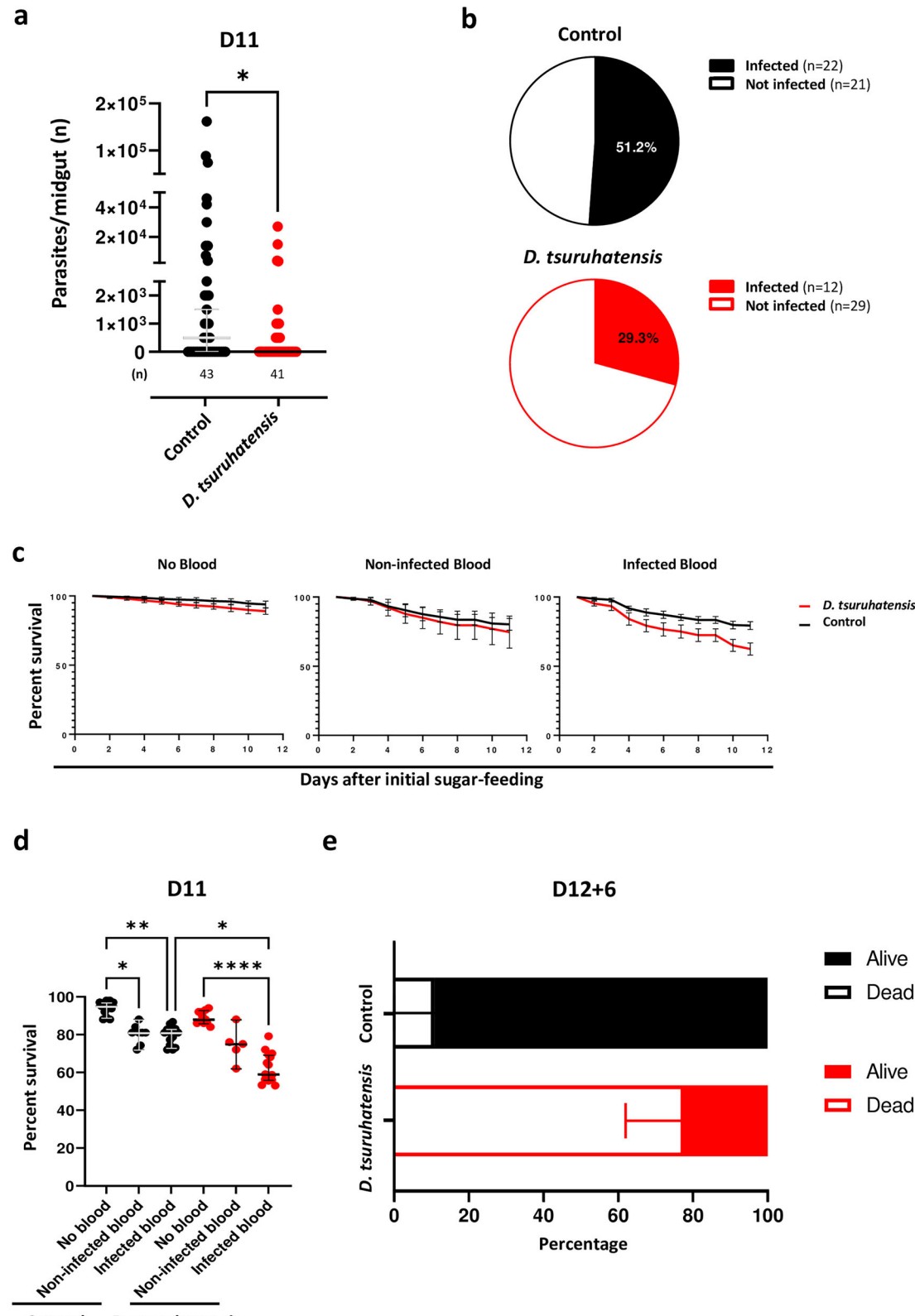

microbiome of wild-caught sand flies, as were multiple other genera detected in the microbiome of our laboratory-reared sand flies[29–33]. Therefore, the dysbiosis-related effect on *Leishmania* within sand flies observed here, should also apply to wild insects. This said, studies with wild-caught sand fly populations are needed to understand if the expected induction of dysbiosis will still promote refractoriness to *Leishmania* spp. infection. Of note, although an inhibition effect of the

bacterial supernatant was not observed in vivo, our data does not exclude the hypothesis of a direct effect of *D. tsuruhatensis* TC1 on the establishment of *Leishmania* in sand flies (e,g. the anti-*Leishmania* component(s) in the bacterial supernatant may be rapidly degraded within the sand fly midgut, and thus, a constant in situ secretion of such component(s) by live bacteria is needed for the successful control of *Leishmania* spp. parasites). Therefore, we may speculate that our

**Fig. 5 | *Delftia tsuruhatensis* colonization reduces both the *Leishmania* infection burden in sand flies and the overall prevalence of infected sand flies.**
**a**, **b** Sand flies took blood from active cutaneous leishmaniasis lesions and then were allowed to feed overnight on sugar containing or not *D. tsuruhatensis*. Eleven days later, the midgut infection burden was quantified microscopically. **a** Total number of parasites per midgut of control (black dots) and *D. tsuruhatensis*-fed (red dots) sand flies. Each dot represents a single midgut. The group median ± 95% CI are shown. Statistical significance was determined using the Mann-Whitney test (two-sided): *p = 0.0167. **b** Prevalence of infected flies in the control (black pie chart), and bacteria-fed (red pie chart) groups. A relative risk of infection of 0.6491 in bacteria-fed versus control sand flies was calculated (95% CI of 0.4277–0.9820; p = 0.0482; Fischer's exact test; two-sided). **c**, **d** The mortality of sand flies that never took a bloodmeal, that took a non-infected bloodmeal, and that took an infected blood-meal was recorded comparing bacteria-fed *versus* control groups. **c** Percentage of surviving sand flies with time in control (black lines) and bacteria-fed (red lines) sand flies given only sugar (n = 12 x 100 flies and n = 11 x 100 flies, respectively; left panel), a non-infected bloodmeal (n = 8 x 100 flies and n = 5 x 100 flies, respectively; middle panel), or an infected bloodmeal (n = 14 x 100 flies and n = 15 x 100 flies, respectively; right panel). Mean, and 95% confidence interval values are represented. **d** Snapshot of the survival data at day 11. Each dot refers to a 100-sand fly group. Bars show the median ± 95% CI. Statistical significance was determined using the Kruskal-Wallis test with Dunn's post-hoc analysis: *p = 0.0187 (Control No blood versus Control Non-infected blood), *p = 0.0383 (Control Infected blood versus *D. truruhatensis* Infected blood), **p = 0.0013, and ****p = 1.7915 × 10⁻⁶. **e** Sand flies were allowed to take a sugar meal alone (control), or containing *D. tsuruhatensis*, infected with *L. major* parasites at day 1, and given a second, non-infected blood-meal at day 12. The overall survival of control (black bar graph) and bacteria-fed (red bar graph) *Leishmania*-infected sand flies, 6 days after ingesting a second, non-infected bloodmeal is represented (n = 6 x at least 10 sand flies per group). Mean, and 95% CI are shown. The relative risk of dying in bacteria-fed versus control sand flies was calculated (7.216; 95% CI of 4.224–12.78; p < 0.0001; Fischer's exact test; two-sided). All results were obtained in, at least, two independent experiments. Source data are provided as a Source Data file.

phenotype can be a consequence of both, a direct effect of bacteria excreted/secreted products on *Leishmania* spp. parasites, and an indirect effect, consequence of *D. tsuruhatensis* TC1-induced sand fly gut dysbiosis.

Importantly, *Delftia* spp. are environmental bacteria, widely distributed in nature[34], and are part of the microbiota of different human-disease vectors, including sand flies, mosquitoes, and ticks[11,35]. Therefore, the use of *D. tsuruhatensis* TC1 for the vector-based control of leishmaniasis is not expected to disturb natural ecosystems. However, the changes in the microbiota of wild hematophagous insects may be associated with risks, particularly when we consider the possibility of indirect pathogenic effects. We know that, together with *Leishmania* parasites, sand flies regurgitate a complex inoculum into the skin[1], including sand fly midgut bacteria[36]. This has the potential to lead to *Leishmania* spp.-bacteria co-infection scenarios that are sporadically reported[37]. Of note, most of these scenarios are probably just a consequence of the progression of leishmaniasis and not of vector bites-derived inoculation; visceral leishmaniasis is known to induce immune suppression[38] and cutaneous leishmaniasis may lead to the compromise of the skin barrier function[39], both risk factors for the establishment of opportunistic pathogens. Of relevance, the inoculation of gut bacteria via sand fly bites is dependent on the infection status; the higher the infection, the greater the probability of regurgitation to allow blood-feeding[13,14]. Therefore, the fact that *D. tsuruhatensis* colonization leads to a decrease in the sand fly *Leishmania*-infection burden, may suggest these sand flies are egesting lower levels of gut bacteria into the host skin, making it less probable for these insects to transmit any resilient bacterial agent that becomes prevalent in a sand fly gut dysbiosis context. Of note, excluding CL lesions, no visible clinical manifestations were observed in our animals, including those exposed to the bites of *D. tsuruhatensis*-colonized sand flies. Adding to this, no *Delftia* secondary infection in the context of leishmaniasis was reported to date, whereas *Serratia* spp.-*Leishmania* spp. co-infection was reported only once[37]. This said, in depth studies on the potential co-transmission of potentially pathogenic bacterial strains with *Leishmania* spp. parasites in the context *D. tsuruhatensis* TC1 colonization need to be carried out to support the development of any mitigation strategies, if needed.

To our knowledge, to date, no single vector-based control approach aiming to promote refractoriness was deemed effective in the field for the simultaneous control of malaria and leishmaniasis (targeting both *Leishmania* spp. parasites within sand flies, and *Plasmodium* spp. parasites within mosquitoes, respectively). For instance, while *Wolbachia* strains have been shown to impact the transmission of flaviviruses and *Plasmodium* parasites by mosquitoes[40,41], the same is yet to be demonstrated regarding *Leishmania* transmission by sand flies[42]. Considering that malaria and leishmaniasis are co-endemic in more than 50 countries worldwide[43,44], our data is of great relevance. Our results regarding sand flies, together with those recently reported with mosquitoes[5], support the development of *Delftia tsuruhatensis* TC1-based interventions for the concurrent control of leishmaniasis and malaria in the field. Of note, while for mosquitoes, the targeting of the larval stage is an option, for sand flies such an approach would be quite challenging. Unlike mosquitoes, sand flies do not have an aquatic larval stage; they lay their eggs in organic matter-rich environments, which are difficult to locate and mostly unknown[1]. Therefore, the reduced potential for bacterial dispersion in soil (versus water) makes deploying *Delftia tsuruhatensis* TC1 in such non-aquatic sites unfeasible. With this in mind, we envision, instead, the use of sugar baits adapted for targeted delivery, in places where the insects are expected to occur (e.g., breeding sites). Importantly, such sugar baits will not only contain the bacterial suspension for delivery, but also specific odorant-attracting molecules to make them more competitive than natural sugar sources (particularly in places where such sources are abundant), and prevent potential sub-optimal deployment, as reported elsewhere in a different context[45]. Of note, several molecules were reported to attract mosquitoes and sand flies, including some that have the potential to attract both insect species simultaneously (e.g., 1-octen-3-ol)[46,47]. The development of these sugar baits is expected to be done in the near future, for use in the context of semi-field studies that will allow us the extrapolation of our results to the field context.

## Methods

### Ethics statement
All animal studies were conducted in accordance with the GSK Policy on the Care, Welfare and Treatment of Laboratory Animals and were reviewed by the National Institute of Allergy and Infectious Diseases (NIAID) Animal Care and Use Committee (under the animal protocol LMVR4E).

### Mice
Six-week-old female BALB/c mice were obtained from Charles River laboratories (catalog # BALB-F) and housed under pathogen-free conditions at the NIAID Twinbrook animal facility (Rockville, MD) at 18–23 °C room temperature, and 40–70% relative humidity, under 12 h dark/light cycles, and with water and food ad libitum.

### Parasites
A cloned line of *Leishmania major* (WR 2885) was used[48]. Promastigotes were maintained at 26 °C in Schneider's insect medium supplemented with 20% heat-inactivated fetal bovine serum, 100 U/mL penicillin, and 100 mg/mL streptomycin (all Thermo Fischer Scientific).

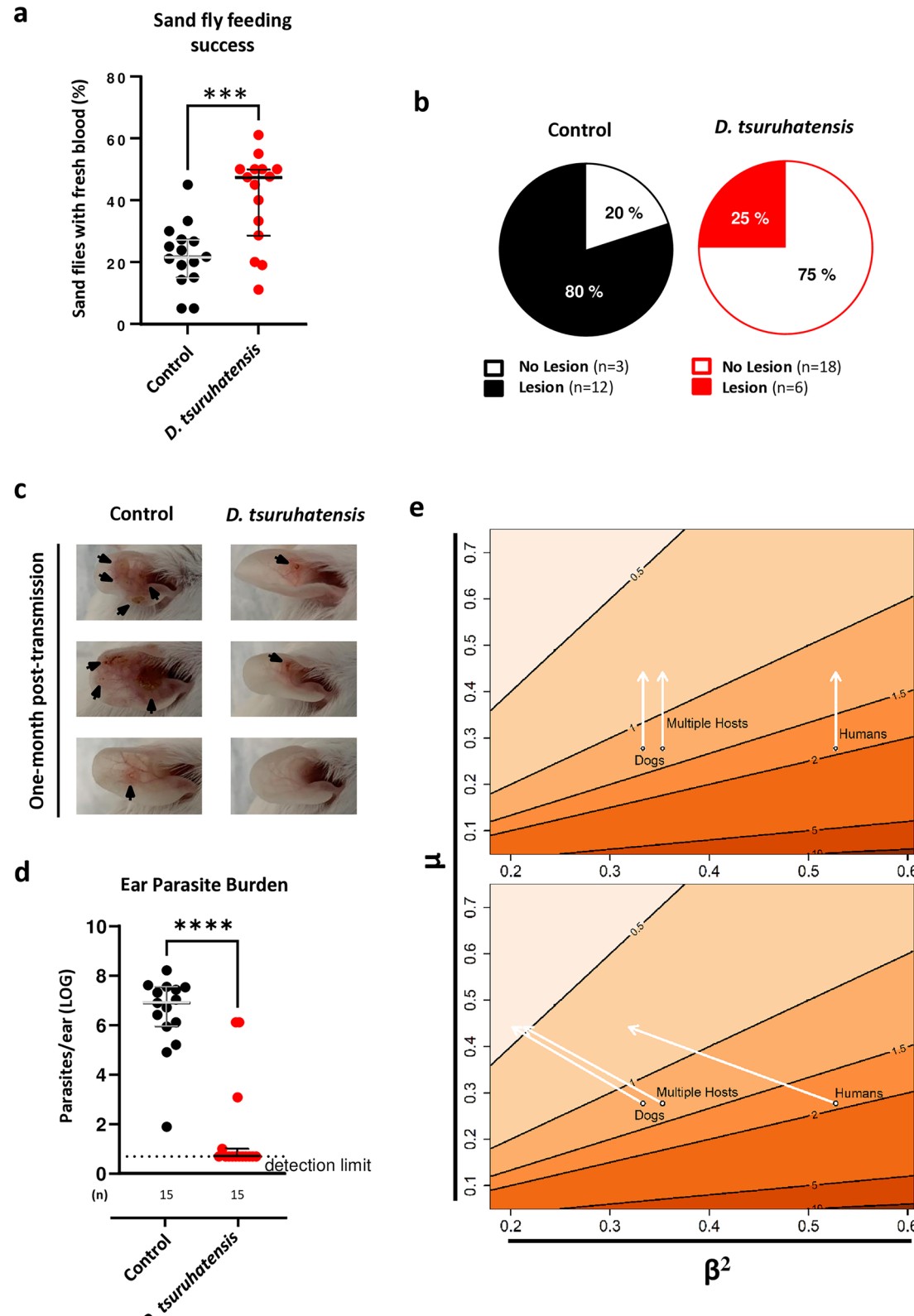

**a** Sand fly feeding success

**b** Control / *D. tsuruhatensis*

Control: No Lesion 20%, Lesion 80%
*D. tsuruhatensis*: No Lesion 75%, Lesion 25%

No Lesion (n=3), Lesion (n=12)
No Lesion (n=18), Lesion (n=6)

**c** One-month post-transmission — Control / *D. tsuruhatensis*

**d** Ear Parasite Burden

**e**

## Bacteria

An *An. stephensi* colony from GSK's insectary at Global Health Medicines R&D site in Tres Cantos, Spain, gradually lost susceptibility to *P. falciparum*, but mosquito survival and reproductive ability were not compromised. Our collaborators isolated a *Plasmodium* refractoriness-promoting bacterium from the midguts of these mosquitoes[5]. Briefly, homogenized tissues from pupae, larvae, adult female mosquito midguts and water from larval breeding pans were transferred to sterile LB broth and incubated at 27 °C or at room temperature on a rotary shaker. When turbidity was observed, samples were plated on solid LB agar to obtain single colonies. Colonies were selected based on distinct morphology, transferred to fresh liquid media and subsequently identified both via biochemical characterization using API strips (Biomerieux) and via 16sRNA typing. A bacterium identified as *Delftia*

**Fig. 6 | *Delftia tsuruhatensis* colonization impairs *Leishmania* transmission by sand flies with relevance for disease control in endemic settings. a–d** Bacteria-fed and control sand flies were infected with *L. major* parasites, kept for 11 days, and then allowed to feed from the ears of BALB/c mice. Animals were followed up for one month and pathological alterations were recorded; ear parasite burdens were finally determined via limiting dilution. **a** Percentage of bacteria-fed (red dots), and control (black dots) sand flies that were able to blood-feed from the ears of BALB/c mice. Each dot represents a group of 20 sand flies. The group median ± 95% CI is shown. **b** Prevalence of animals developing active lesions after being bitten by control (black pie-chart), or bacteria-fed (red pie-chart) *Leishmania*-infected sand flies. The relative risk of disease development in mice bitten by bacteria-fed versus control sand flies was calculated (RR = 0.2361; 95% CI of 0.08004-0.6084; p = 0.002; Fischer's exact test; two-sided). **c** Pathologic alterations in animals bitten by control (left panel), or bacteria-fed (right panel) *Leishmania*-infected sand flies. **d** Ear parasite burdens of animals bitten by control (black dots), or bacteria-fed (red dots) *Leishmania*-infected sand flies. Each dot represents one ear. The group median ± 95% CI is shown. Statistical significance was determined using the Unpaired t-test (**a**; two-tailed) or the Mann-Whitney test (**d**; two-tailed): ***$p$ = 0.0005 and ****$p$ = 1.34092 × 10$^6$. All results were obtained in, at least, two

independent experiments. **e** Basic Reproduction Number ($R_O$) surfaces were generated as function of the infection rate ($\beta^2$) and 14-day mortality ($\mu$) for a leishmaniasis transmission model with reservoirs/incidental hosts[75,77], as per the equation $R_0 = C\frac{\beta^2}{\mu}$ where $C$ is a parameter that is a function of reservoir host and vector relative abundance, which we set at $C = 2$ in the plot. On the surface, black circles represent $R_O$ estimates from three datasets, a cross sectional serosurvey in dogs[15] (Dogs, $\hat{R_O} = 1.20$), an outbreak investigation with multiple vertebrate hosts/vectors[17] (Multiple Hosts, $\hat{R_O} = 1.27$), and the analysis of a time-series of human cases[18] (Humans, $\hat{R_O} = 1.90$). Diagonal lines represent parameter combinations with the same R0 value, while different shades of orange represent ranges between the values demarked by the diagonal lines. To locate $R_O$ estimates on the surface we assumed infection and mortality as determined for our experimental controls ($\hat{\beta}^2 = 0.51$ and $\hat{\mu} = 0.28$). White arrows represent the reduction in $R_O$ when parameters changed based on the treatment results from our experiments, with mortality increasing alone ($\hat{\mu} = 0.44$; top panel), or together with reduced transmission potential (bottom panel; $\hat{\beta}^2 = 0.33$). $R_O$ estimates were reduced as follows: Dogs ($\hat{R_O} = 0.75$, or $\hat{R_O} = 0.45$), Multiple Hosts ($\hat{R_O} = 0.80$, or $\hat{R_O} = 0.48$) and Humans ($\hat{R_O} = 1.19$, or $\hat{R_O} = 0.72$). Source data are provided as a Source Data file.

*tsuruhatensis* was found to be predominantly present in all screened samples and the strain was designated as *Delftia tsuruhatensis* Tres Cantos 1 or TC1. A GFP-expressing *Delftia tsuruhatensis* was further generated and used for imaging purposes.

### Sand flies

*Phlebotomus duboscqi* sand flies were mass-reared at the Laboratory of Malaria and Vector Research insectary as previously described in ref. 49. Adult females were maintained on a 30% sucrose diet and were starved for 12 h before feeding.

### Bacterial growth and preparation

*D. tsuruhatensis* was grown in LB liquid medium containing 50 μg/ml carbenicillin (both from Sigma-Aldrich, St. Louis, MO, USA) overnight at 28 °C under aeration conditions (250 rpm). The culture was then centrifuged at 4,000 × g for 5 min and washed 3 times with PBS. For bacterial feeding experiments the number of colony-forming units (CFUs) was adjusted via the measurement of the optical density (600 nm wavelength). The calculated volume was then re-suspended in sterile 5% sucrose solution. For experiments with killed bacteria (fig. S5), the bacterial suspension was incubated at 95 °C for 10 min prior to dilution in sucrose solution.

### Bacterial supernatant preparation and fractionation

The bacterial excreted-secreted products were obtained as described elsewhere[5]. Briefly, *D. tsuruhatensis* TC1 bacteria were grown in LB liquid medium overnight (200 rpm, 28 °C), washed with PBS, resuspended in M9 medium (10$^9$ CFUs/ml), and incubated (200 rpm, 28 °C) for 8 h. Afterward the TC1 supernatant was collected via a centrifugation step, and passed through a 0.22 μm filter. M9 culture medium without bacteria was subjected to exactly the same conditions and steps for the generation of the M9 control supernatant. For some experiments, we fractionated both the TC1 and M9 supernatants using sequential centrifugation steps in the context of Amicon Ultra ® Centrifugal filters (100 and 10 kDa cutoffs; Millipore Sigma, Burlington, MA, USA).

### Sand fly bacterial (byproducts) feeding

*P. duboscqi* sand flies were allowed to feed on cotton rolls impregnated with a suspension of TC1 bacteria (WT or GFP-expressing; alive or dead), or *E. coli*, or an *Ornithinbacillus massiliensis* strain directly isolated from the midguts of sand flies (1 × 10$^8$ CFU/ml in 5% sucrose solution) for 24 h; sand flies in the control group were given 5% sucrose alone. Bacterial feeding was done either one week prior to infection, 24 h prior to infection, 5 days post-infection, or 8 days post-infection. The bacterial supernatant was also given to sand flies in the same

manner. In this case, sucrose was directly diluted into both TC1 or M9 control supernatants which served as the sugar vehicle. For these experiments, the supernatant was given to sand flies on a daily basis, after infection.

### Sand fly infection

After an overnight starving period, sand flies were infected by artificial feeding through a chick membrane on defibrinated rabbit blood (Spring Valley Laboratories, MD, USA) containing *L. major* promastigotes (5 × 10$^6$/ml), as previously described in ref. 50. In some instances, 50 μM harmane (Sigma-Aldrich, St. Louis, MO, USA), or 50% (bacterial) supernatant were added to the infected blood mixture. After infection, blood-fed females were sorted and kept on a 30% sucrose diet. At 7-, and 11/12-days post infection sand flies were collected to assess the infection status. The number of parasites was determined under the optical microscope using a Neubauer chamber, and the thickness of the anterior midgut was measured under a Leica DFC 7000 T stereomicroscope using the software Leica applications suite X, v3.7.5.24914. Additionally, at 11 days post-infection, sand flies were used for transmission experiments.

### Pick up experiments

BALB/c mice were infected intradermally in the ear pinnae with 1000 *L. major* metacyclic parasites and kept with water and food ad libitum until the development of typical CL lesions. Then sand flies were allowed to take a bloodmeal from the ears with active lesions using vials with a meshed surface held in place by custom-made clamps. Blood-engorged sand flies were separated into 2 groups. One group was then allowed to feed on the bacteria via sugar-meal overnight (1 × 10$^8$/ml in 5% sterile sucrose solution), while the other received 5% sterile sucrose solution alone. Sand flies from both groups were dissected 11 days post-infection to evaluate their infection status.

### Non-infected second bloodmeal

After the sand flies had taken an infected blood meal, either by feeding on a glass feeder or an infected animal, they were kept on a 30% sucrose diet for 12 days. The sand flies were then allowed to blood engorge on an anesthetized naïve mouse for one hour, as reported elsewhere[51]. Blood-fed females were sorted and kept on a 30% sucrose diet, and 6 days post feeding sand flies were collected to assess the infection status.

### Sand fly infection status evaluation

Under a stereomicroscope, sand fly midguts were dissected in PBS and transferred to individual microtubes (Denville Scientific) with 50 μL of

formalin solution (0.005% in PBS). Midguts were homogenized and 10 µL loaded onto disposable Neubauer chambers (Incyto). Slides were observed under a phase contrast microscope (Zeiss) at 400× magnification. The total parasite numbers, and the metacyclic frequency[48] were determined.

### Fluorescence imaging

Under a stereomicroscope, sand fly midguts were dissected in PBS, over a glass slide, and covered with a coverslip. Phase-contrast and epi-fluorescence imaging was carried oved immediately after dissection. Only intact midguts were used for image acquisition. Images were taken using a K5 camera coupled to a Thunder Imager microscope (Leica Microsystems, Wetzlar, Germany) using the Leica Application Suite X software platform. Images were processed with the Imaris software v10.2.0 (Oxford Instruments, Abingdon, UK). Images were exported using the Imaris snapshot tool. If present, the grey contour around an image is from the software display screen, which is kept for frame size uniformity. Images were pseudo-colored since the camera is a black and white one, for increased sensitivity. Histograms were adjusted equally for the channels across all images. All raw images are available upon request.

### Metagenomics analysis – layout and samples

Control or *D. tsuruhatensis*-fed bacteria sand fly midguts were dissected under a sterile-like environment at different times post bacterial/blood feeding: after bacterial feeding but before blood feeding (D0), and after bacterial and blood feeding (D2, D5, D7, D9, and D12). Dissected midguts were washed three times in sterile PBS drops and then transferred to an Eppendorf containing 50 µL of sterile PBS; pools of 20 midguts were collected per condition at least in triplicate. The midgut pools were then macerated using a motorized pestle (Kimble Chase, Vineland, New Jersey). Genomic DNA was finally extracted using the EZNA Tissue DNA Kit (D3396-01; Omega Bio-Tek, Norcross, GA, USA) according to the manufacturer's instructions, and the samples were subjected to 16S rRNA amplification and sequencing as reported elsewhere[11]. Overall, a total of 42 samples were analyzed. For the analysis, frequently, samples were grouped by time-point.

### Metagenomics analysis - amplicon sequence variant calling, phylogenetic tree, and taxonomy classification

A total of 7,758,941 raw reads were generated. 16S rRNA amplicon reads were demultiplexed and trimmed using Novogene in-house scripts. The trimmed demultiplexed reads were then imported into QIIME2 version 2021.4[52] for downstream analysis. DADA2[53] was used to call amplicon sequence variants (ASVs). Chimeric sequences were identified and removed by setting the flag "--p-min-fold-parent-over-abundance" to 10, which discards chimeric sequences showing an at least ten times lower abundant than their parent sequences. After DADA2 quality filtering, a total of 6,911,632 reads (164,562 ± 4,808 reads/sample) were retained and 3735 ASVs were called. A rooted phylogenetic tree was generated using FastTree[54] based on ASV multiple alignment with MAFFT[55]. The ASVs were taxonomically classified with a Naïve Bayes classifier pre-trained on SILVA rRNA database (release 138 SSURef NR99)[56]. All code used to do the metagenomics analysis is fully available[57].

### Metagenomics analysis - microbial diversity calculation, statistical testing, and differential abundance testing

The diversity metrics were calculated using the QIIME2 core-metrics-phylogenetic function after rarefying the feature table to a subsampling depth of 149,894 reads/sample. All 42 samples were retained after rarefaction. The rarefied feature table was used to compute Shannon's diversity metrics[58], observed features, Faith's phylogenetic diversity index[59], Pielou's evenness index[60], and UniFrac distance[61]. To statistically compare each alpha diversity metrics between groups, Wilcoxon rank-sum tests[62] were applied to compare group median values. To visualize

the dissimilarities in microbial communities across groups, the weighted and unweighted UniFrac distance metrics was used to generate PCoA coordinates after Cailliez transformation[63] to correct for negative eigenvalues. Permutational multivariate analysis of variance (PERMANOVA)[64] and permutational multivariate analysis of group dispersion homogeneity (PERMDISP)[65] were applied to compare the centroid location and within-group dispersion level, respectively, in UniFrac distance metrics across groups. Analysis of compositions of Microbiomes with Bias Correction (ANCOM-BC)[66] was also used to find differentially abundant genera between conditions on each day. The unrarefied features table was used as input to ANCOM-BC as per the standard recommendations. All code used to do the metagenomics analysis is fully available[57].

### Shotgun metagenomic sequencing - data preprocessing, assembly, binning, and classification

To reach bacterial identification at the species level when applicable, we conducted shotgun metagenomic sequencing. The quality check, trimming and human contamination removal were performed using the MetaWRAP[67] read_qc module. All samples were pooled and co-assembled using metaSPAdes[68]. The contigs were then binned using the metaWRAP binning module, based on a combination of CONCOCT[69], maxbin2[70], and MetaBAT[71]. The draft bins were then consolidated using the metaWRAP bin_refinement module that merges bin sets generated from the three binning algorithms into one set of de-replicated and improved bins. A total of four high quality metagenome-assembled genomes (MAGs) were acquired, each with a completeness > 98% and a contamination level < 2.2%. The MAGs were then taxonomically classified with GTDB-Tk v2[72] (version 2.1.1) using the "--full_tree" mode. GTDB-Tk provides accurate taxonomic assignment of MAGs based on the concordance of placement of MAGs in the GTDB reference phylogenetic tree and the average nucleotide identity to representative reference genomes. The abundance of MAGs in samples were quantified using the metaWRAP quant_bins module. Particularly, within the module, salmon[73] was used to estimate the coverage of each contig in every sample; the coverage value of contigs within each bin was then length-weighted averaged to estimate the mean abundance of each bin in each sample. All code used to do the metagenomics analysis is fully available[57].

### Leishmania in vitro inhibition assays

*Leishmania major* were "seeded" into 24-well plates in Schneider's insect medium supplemented with 10% heat-inactivated fetal bovine serum, 100 U/mL penicillin, and 100 mg/mL streptomycin (all Thermo Fischer Scientific) at the density of 1 to 2 x 10^5 parasites/mL. Increasing concentrations of TC1 supernatant or harmane were added to the wells; in parallel, the respective amount of M9 supernatant or DMSO vehicle were added to control wells. In some instances, TC1 and M9 supernatant fractions were also added at a final concentration of 50%. The parasite numbers were counted every day up to seven days after plating.

### Sand fly fitness evaluation

Bacteria-fed or control sand flies, not fed on blood, artificially fed on non-infected blood, or artificially fed on blood containing *L. major* promastigotes (5 x 10^6/ml) were placed into cardboard pints (100 flies per pint) and kept on a 30% sucrose diet for 11–18 days. Sand fly mortality was recorded daily, and the dead sand flies were removed also daily. Additionally, for the infected sand flies only, at day 12 post-infection, a second bloodmeal was given (as above described) and the mortality was also recorded daily only considering the sorted blood fed sand flies in the following 6 days.

### Transmission of L. major via sand fly bites

Only blood fed females were used in transmission experiments. At day 11 post feeding, 20 sand flies were applied to each mouse ear, using vials with a meshed surface held in place by custom-made clamps, and

allowed to feed. The number of blood-fed flies was determined by observing them under a stereomicroscope.

## Post-transmission follow-up and experimental endpoints

Animals were monitored on a weekly basis to follow the development of lesions caused by *L. major* infection; images of individual ears were captured weekly using a smartphone. Animals were euthanized 4 weeks post infection for parasite burden determination.

## Parasite burden determination

Euthanasia was performed by cervical dislocation under isoflurane anesthesia in all cases (Piramal Healthcare). Ears were collected, immersed in 70% ethanol and then in PBS. Ear cell suspensions were obtained as previously described in ref. 74. Ear parasite burdens were assessed by the limit dilution method. Briefly, ear cell suspensions were serially diluted in 96-well plates containing 200 μL Schneider's insect medium supplemented with 20% heat-inactivated fetal bovine serum, 100 U/mL penicillin, and 100 mg/mL streptomycin (all Thermo Fischer Scientific), and incubated for 14 days at 26 °C. The total number of parasites per ear was calculated as previously[50].

## Modelling

To explore the potential impacts of the bacterial infection on *Leishmania* spp. parasite transmission we used analytical results from a mathematical model that we developed to study the dynamics of leishmaniasis transmission by vectors in a community of two vertebrate hosts[75], where one is a reservoir host while the other is an incidental host (e.g. a host that only gets infected but cannot infect sand fly vectors). In this model the basic reproduction number ($R_O$, the average number of new infections in a susceptible population following the introduction of an infected host[76]), is defined by the following equation:

$$R_0 = \frac{\beta^2 BV}{\mu \gamma} \tag{1}$$

where $V$ is the population size of vectors, $B$ the population size of reservoirs, $\gamma$ the recovery rate of reservoirs, $\mu$ the mortality rate of vectors, and $\beta$ the transmission rate, which is an abstraction of the processes that following the contact between vectors and reservoirs can lead to infections in an uninfected host.

To assess the impact of the bacteria in transmission first we assume that $C = \frac{BV}{\gamma}$ is a constant that doesn't change when the bacteria is introduced in an ecosystem with *Leishmania* spp. transmission. Under this assumption Eq. (1) becomes:

$$R_0 = C\frac{\beta^2}{\mu} \tag{2}$$

Here it is important to highlight that $R_0 \propto \frac{\beta^2}{\mu}$ when more than 2 vertebrate host species are present[77]. Therefore, the use of Eq. (2) is better suitable when we do not know exactly the vertebrate host species diversity in a transmission context (which is often the case thinking on leishmaniasis). Furthermore, with Eq. (2) we can then explore how $R_O$ changes in response to changes in $\mu$ and $\beta$. For $\mu$ this can be done by assuming that in a natural transmission system, $\mu$ is given by the survival of sand flies in our control group. We can then estimate the change in $R_O$ assuming that changes in sand fly mortality, $\mu^{\#}$, will be the same in the field as the ones observed in the laboratory when sand flies are colonized with *D. tsuruhatensis*. From Eq. (2) we can expect that if $\mu$ increases when sand flies are colonized with the bacteria (particularly *Leishmania*-infected sand flies), $R_O$ should decrease. For $\beta$, we can assume that its magnitude will change following proportional changes in the dissemination of *Leishmania* spp. infections. After introducing *D. tsuruhatensis*, we can then assume that $\beta^{\#} = \beta^2 \frac{SF+}{SF-}$, where SF+ is the proportion of disseminated *Leishmania* spp.

infections in vectors when bacteria are present, while SF- is the proportion of the disseminated infections in the absence of the bacteria. Thus, the basic reproduction number after the introduction of the bacteria, $R_0^{\#}$, becomes:

$$R_0^{\#} = C\frac{\beta^{\#}}{\mu^{\#}} \tag{3}$$

Using Eq. (3), we explored changes in $R_O$ using baseline estimates from the field, including a study where $R_O$ was based on a cross-sectional study of *Leishmania* spp. exposure in dogs from Panama[15], a multi-species leishmaniasis outbreak from Venezuela[17,78], and the analysis of a time series of human leishmaniasis cases from Costa Rica[18]. In the supplementary materials we include the code used to illustrate how changes in $\beta$ and $\mu$ can lead to changes in $R_O$ by plotting changes over a surface that represents the relationship of $R_O$ to $\beta$ and $\mu$. Additionally, the parameter estimates used in our modelling approach are also listed in our supplementary materials, in Table S2.

## Data representation and statistics

Results obtained in at least 2 independent experiments are shown per individual sand fly/mouse, with a representation of the group mean/median value ± 95% confidence interval (CI), unless otherwise stated. Statistical analysis was performed using GraphPad Prism software v6.01. All datasets were first subjected to normality tests (Shapiro-Wilk or Kolmogorov-Smirnov). The Unpaired t-test (parametric; always two-tailed) was used to access statistical differences when all groups in a dataset showed normal distributions. The Mann-Whitney test (non-parametric; always two-tailed) or Kruskal-Wallis test, the latter with post-hoc analysis (Dunn's test), were used to access statistical differences when at least one group in a dataset did not show a normal distribution. A *p*-value ≤ 0.05 was considered statistically significant.

## Reporting summary

Further information on research design is available in the Nature Portfolio Reporting Summary linked to this article.

## Data availability

The bacterial strain *D. tsuruhatensis* TC1 is available from the NCIMB using the accession number 43398 under a material transfer agreement with NCIMB. The metagenomics raw data generated in this study have been deposited in the NCBI GeneBank database, under the BioProject number: PRJNA1079352. All remaining data are available in this manuscript and its supplementary materials, as well as in the form of Source Data, provided with this paper. Source data are provided with this paper.

## Code availability

The code used to analyze the metagenomics data (including the shotgun sequencing analysis that led to the identification of *S. ureilytica*) is available at https://github.com/GaryZhangYue/Cecilio_2024_TC1_sandflies[57]. The code used for the modelling is provided as the Data S2 file.

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

## Acknowledgements

We thank Dr. Jose M.C. Ribeiro for the fruitful discussion and his feedback on the manuscript. This work was supported by the Intramural Research Program of the National Institute of Allergy and Infectious Diseases, National Institutes of Health. The funder had no role in study design, data collection, data analysis, data interpretation, writing of the manuscript, and decision to publish.

## Author contributions

P.C. and F.O. conceived the study and designed the experiments. P.C., L.A.R., T.D.S., K.T., L.W., E.I., Ld.S.F., M.G.G. and F.O. performed the experiments and analyzed the data. L.F.C. did the modelling. Y.Z. did the metagenomics analysis. J.G.V. and F.O. assured the funding. C.M., W.H., P.C.C., M.J.L., J.G.V., and J.R. contributed with reagents, materials, and analysis tools. P.C. wrote the original draft. All authors critically discussed the results, and revised, edited, and approved the manuscript.

## Funding

## Competing interests

The data and research presented in this manuscript are disclosed in the international patent with the publication number WO2024/100131 pertaining to the author Janneth Rodrigues. The remaining authors declare that they have no competing interests.
