## [Peer Review file · Nature Communications]

Leishmania sand fly-transmission is disrupted by *Delftia tsuruhatensis* TC1 bacteria

Corresponding Author: Dr Fabiano Oliveira

Version 0:

Reviewer comments:

Reviewer #1

(Remarks to the Author)

Key results

The authors found that treatment of female *P. duboscqi* with *Delftia tsuruhatensis* caused a reduction in the number of *L.* major promastigotes in their guts. The effect was induced only by treatment with live bacteria; the culture supernatant did not affect the infection and the specificity of the effect was confirmed by experiments with other bacterial species (*E. coli*, *O. massiliensis*) and dead *D. tsuruhatensis*, where *Leishmania* infection was unaffected. *Delftia tsuruhatensis* did not alter the alpha diversity of the bacterial community in the gut of *P. duboscqi*, but did affect the relative abundance of different bacterial genera. Specifically, it induced a dominance of the genus *Serratia*, harming *Leishmania* infection. I appreciate that the authors tested the effect of *D. tsuruhatensis* treatment not only before leishmania infection but also in females infected first by *Leishmania*. And valuably, they included experiments where female sand flies were infected naturally by feeding on the host: the difference between the control and treated groups was confirmed even under this arrangement. In terms of potential use in the wild, it is remarkable that significantly higher mortality of sand flies compared to controls was observed only in infected blood fed females. The reduction of *Leishmania* transmission to the host in the treated group of sand flies is impressive; moreover, infected mice were often asymptomatic. Finally, the authors also used a mathematical model to verify that colonization of sand flies by *D. tsuruhatensis* has the potential to disrupt transmission in endemic sites.

Validity and analytical approach

The vast majority of the results are valid, well analysed and interpreted. Data are mostly presented in graphs; sample size and the statistical tests used are always indicated. The only missing test of statistical significance concerns data summarized in the graphs of Fig. 4B, Fig. 5B (difference between the control and treatment groups in the prevalence of infected flies and mice that developed lesions, respectively).

Methodology

The authors used adequate methods that are sufficiently described in the supplementary materials. I just have a comment on the evaluation of sand fly infections: the effect of *D. tsuruhatensis* on *Leishmania* infection is presented in terms of the number of promastigotes in guts. I am surprised that there is no assessment of the prevalence and localization of infections; I only found the prevalence of infections initiated by feeding on the host. Could authors provide comparison of percentage of infected females in treated and control groups for artificial membrane feedings?

Regarding parasite localization, it is important to know whether promastigotes colonize the stomodeal valve in mature infection. I like to ask authors to show whether the proportion of females with colonized stomodeal valve differed between the control and treatment groups. However, if the authors do not have these data, I do not insist on completing the additional experiments because transmission to mice was demonstrated.

Significance

This work represents original research with stimulating implications for leishmaniasis control. It shows that *Delftia tsuruhatensis*, which has previously been used successfully in *Anopheles* mosquitoes to reduce their ability to transmit *Plasmodium*, is also effective in reducing *Leishmania* transmission by sand flies under laboratory conditions. We can only wish that this efficacy is confirmed in the field. However, it is also important to point out the safety risks associated with this control application, which I recommend to include in the discussion.

Clarity and context

The text is very clearly written, and the chapters with results are logically linked. The Introduction and Discussion are not too lengthy, yet they cover important facts and discuss relevant previous results by other authors. However, it would be helpful to add a few points (specified below) to the discussion.

I have some minor comments in the following paragraph:

- Abstract should contain no references.
- Line 56 – reference number 8 is not optimal, an older publication like Maroli 2012 (doi: 10.1111/j.1365-2915.2012.01034.x) should be cited.
- I propose to integrate the results of the chapter Specificity of the *Delftia tsuruhatensis* effect on *Leishmania* infection (Fig. 1 and Fig. S1) into a single figure. The scheme in Figure S1 could be simplified, e.g., by omitting the image of the dissected sand fly.
- Lines 112-113: If there is a single species of the genus *Serratia* (*S. ureilytica*), the abbreviation *Serratia* sp. should be used. The abbreviation *Serratia* spp. would correspond to a situation where there are multiple species in the sample.
- Lines 263-264: "To our knowledge, no vector control approach aiming to promote refractoriness was deemed effective in the field for both malaria and leishmaniasis." Are the authors referring to the same product that would be universally effective against both *Plasmodium* in mosquitoes and *Leishmania* in sand flies? It should be worded more clearly.
- In Discussion, authors explain the observed phenotype by dysbiosis of sand fly guts. What is the authors' opinion on the possible influence of the sand fly immune response on the observed effect? For example, it has been described that defensins are upregulated in *Leishmania* infected *P. papatasi* (10.3390/microorganisms9112307).
- In Discussion, I would appreciate an expanded passage on the potential risk of introducing the model in the field. Since the composition of the microbiome of wild sand flies is variable and differs from that of a laboratory colony, it is difficult to predict what changes in the composition of the bacterial community will occur after *Delftia tsuruhatensis* treatment. A bacterial species may be selected that will support the development of *Leishmania* (e.g. 10.1128/mBio.01121-16, 10.1111/cmi.12755) or species pathogenic to humans. For example, a new opportunistic pathogen, *Serratia marcescens*, exhibiting resistance to many antibiotics (10.3390/antibiotics12020367, 10.7717/peerj.14399) has been described from wild sand flies (10.1155/2020/5458063, 10.1371/journal.pone.0050259).

(Remarks on code availability)

Reviewer #2

(Remarks to the Author)

After carefully reading the manuscript from Cecilio et al., the study appears to be scientifically well designed, and the data generated looks well analyzed and very interesting. The discussion around those data is well elaborated, although one of my key points of suggestion is about the latter part of the discussion.

My minor comments are below:

Line 1: In the title, I suggest adding what TC1 symbiotic bacterium is or something that could give more details instead of just saying *Delftia tsuruhatensis* TC1.

Lines 269-272: The use of sugar baits adapted for targeted delivery of TC1-based strategies needs to be discussed more. A recent study with malaria mosquitoes showed that these sugar bait strategies did not work very well with ATSB in Zambia: Wagman et al. 2024, "Entomological effects of attractive targeted sugar bait station deployment in Western Zambia: vector surveillance findings from a two-arm cluster randomized phase III trial." Considering that most malaria-endemic countries in Africa have humid tropical weather with a lot of nectar naturally available for mosquitoes, this makes sugar baits less competitive than natural sugar sources. This should be discussed, focusing on the fact that sugar baits could work in environments like Sahelian malaria-endemic countries such as Mali, where attractive sugar baits seem to work. In other contexts, authors could think about other strategies, like larval breeding seeding with TC1 or releasing some fraction of TC1-infected sandflies to be disseminated. This delivery component is crucial for the deployment of TC1-based strategies for sandfly control and even for mosquitoes as well

(Remarks on code availability)

NA

Reviewer #4

(Remarks to the Author)

This study demonstrates that *Delftia tsuruhatensis* TC1 can colonize the midgut of sandflies without affecting their growth and development. Meanwhile, *Delftia tsuruhatensis* TC1 can specifically affect the growth and development of *Leishmania* and reduce their transmission ability, effectively reducing the probability of *Leishmania* outbreaks. The research approach is novel, the results are practical and feasible, and can be accepted after revisions.

Here are my suggestions:

Q1: Why choose *Delftia tsuruhatensis* TC1? How effective is *Delftia tsuruhatensis* TC1 in killing *Leishmania* compared to

other bacterial strains or drugs? Comparison is suggested to be made.

Q2: How did the author separate *Delftia tsuruhatensis* TC1? Suggest providing detailed supplements in the part of methods and material.

Q3: What is the concentration of *Delftia tsuruhatensis* TC1 in the study? Is the disruption of *Leishmania* transmission by *Delftia tsuruhatensis* TC1 concentration-dependent?

Q4: Why choose *Escherichia coli* for specific research? I think using only one experiment with *Escherichia coli* cannot draw a conclusion on whether it is specific or not, and it does not have universal significance. More bacterial strains experiments are expected to demonstrate the specificity of *Delftia tsuruhatensis* TC1. And how to determine if *Escherichia coli* has successfully colonized?

Q5: How to determine if *Delftia tsuruhatensis* TC1 has died in specificity experiments? What processing method led to the death of *Delftia tsuruhatensis* TC1? Is the supernatant of dead *Delftia tsuruhatensis* TC1 used in the experiment or bacterial cells?

Q6: The supernatant and bacterial derivatives do not affect the growth of *Leishmania* parasites, indicating that live *Delftia tsuruhatensis* TC1 are the ones that are affected. So, how does live *Delftia tsuruhatensis* TC1 affect the growth of *Leishmania*?

Q7: It is best to indicate the number of experimental subjects in each group in each figure. Why is the number of repetitions different for each group in Figure 2D?

Q8: Will the colonization of *Delftia tsuruhatensis* TC1 affect the vitality and growth of sandflies?

Q9: What are the effects of gut microbiota disruption caused by *Delftia tsuruhatensis* TC1 colonization on the growth and development of sandflies? What impact does it have on the growth of *Leishmania*? Why measure the metagenome of the midgut microbiome? What is the meaning?

Q10: It may be considered to conduct field application experiments of *Delftia tsuruhatensis* TC1 in the wild or in parasitic-endemic areas to observe whether it can reduce the infection rate of *Leishmania*.

Q11: The study did not mention the principle and mechanism by which *Delftia tsuruhatensis* TC1 disrupts the transmission of *Leishmania*. From the author's perspective, what could be the potential mechanisms and principles? Suggest adding and discussing the limitations of this study in the discussion.

(Remarks on code availability)

Reviewer #5

(Remarks to the Author)

This manuscript describes the exploration of *D. tsuruhatensis* infestation as a method of inhibiting the development of *Leishmania* species in sandflies. The authors note that this hasn't been explored before for sandflies but has been shown to be effective in mosquitoes. In this regard the results presented are novel and will be of significance in the field and have the potential for substantial global impact on public health. The paper is well laid out and easy to read. The methodology for the laboratory work seems to be complete and reproducible but this is not my area of expertise. The methodology for the modelling is less clear in the text and supplementary material and some clarification around the statistics is required but otherwise we believe the methodologies are sound and the results support the conclusions made in the paper. With the below (minor) adjustments the paper will be ready for publication.

1. Statistical significance: The manuscript text describes "significant" differences (sometime reductions/sometimes changes), but does not state clearly the p-value at which they consider significance. This seems to be 0.05. This should be stated in the text and consistently in figure captions (it currently differs figure to figure). The authors also need to be clear about what tests were used where. That is, Kruskal Wallis or Mann Whitney and what the alternative hypothesis were (i.e. two-tailed or one-tailed tests). This is technically different between a statistically significant 'difference' and a statistically significant increase/decrease. They also need to justify why they've used different tests.

2. Mathematical model: The mathematical model isn't specified at all in the main text of the manuscript, nor is the appropriate reference to the model source supplied. This is supplied in the supplemental material, but although the R code runs it is difficult to see where the values used actually come from (they are not calculated but are hard coded to produce the figures). The model and the results need to be clarified so that the model based conclusions are properly supported.

Specific comments:

P4L69: "significant reduction" The authors later state that this is a Mann-Whitney test. Is the test two sided? If so it is a significant change. The test itself hasn't tested for reduction/increase. At what level of significance?

P8L173: Is it worth discussing what effect increased mortality has on control programs? Will this mean that the control needs to be re-introduced (i.e. infected sandflies die off so the infection isn't self-sustaining).

P10L206: where does the model of R_0 come from? Is it realistic? What are the assumptions/limitations. This 'model' needs to be better described here.

P11L224: "presumably low ecological impact". This links to comment P8L173. If you are changing their mortality rate can that have a flow on impact?

Figure 1 C and D (and others): The confidence intervals are hidden by the points. It may be worthwhile to change the

confidence intervals to a shade of grey and place them on top of the points for visibility. An alternative may be violin plots which would give the same effect as the jittering of points.

P20L500 (and other figure captions): It would be a good idea to have the ns >0.05; * <0.05 included in the captions and specified in the text for consistency.

Figure 2C and Figure 4D and captions: It's not clear what categories are being tested and which test is being conducted.

(Remarks on code availability)

Most of the code is made available at a github repo. One of us has looked over the code, but have no experience with the specific software used apart from R scripts. Data files are not part of the git repo, so code could not be effectively re-run as far as I could tell. R code for the model of R0 is provided in the supplementary and runs to reproduce the R0 figure (figure 53E). However, values for the plots are hard coded and it isn't always clear where the values come from.

Reviewer #6

(Remarks to the Author)

(Remarks on code availability)

Version 1:

Reviewer comments:

Reviewer #1

(Remarks to the Author)

I appreciate the effort authors made on the manuscript; they considered all the comments and revised carefully the text. I recommend the manuscript for publication in Nature Communications.

(Remarks on code availability)

Reviewer #2

(Remarks to the Author)

After carefully reviewing the revised version of the manuscript by Cecilio et al., I note that all my comments and suggestions have been addressed. The study appears to be scientifically well-designed, and the data generated is well-analyzed and highly interesting. The discussion surrounding the data is also well-articulated.

(Remarks on code availability)

Reviewer #4

(Remarks to the Author)

After reviewing the revised manuscript submitted by the author, I believe that the author has carefully addressed the issues and suggestions I raised. I believe that the author's revised manuscript has met the standards for publication. Therefore, I agree and recommend accepting the publication of this article.

(Remarks on code availability)

Reviewer #7

(Remarks to the Author)

As requested by the editor, I have reviewed the statistical and modelling methods of this paper. The authors have addressed most of the corrections suggested by Reviewer #5; however, I recommend further revisions to ensure the manuscript is suitable for publication:

- Apply one-tailed versions of the non-parametric tests or correct the current interpretation of two-tailed tests.
- Enhance the legend of Figure 6e for better clarity.
- Correct a minor typo in the R code.

Please read below for more details:

1) Statistical significance tests

Reviewer #5 raised a valid point regarding the need for additional details on the Mann-Whitney and Kruskal-Wallis tests. The authors have addressed this concern in the current version of the manuscript, making it clearer where non-parametric tests were applied. However, the Methods section could still be enhanced by providing a rationale for selecting these tests over others, such as the t-test, particularly in the context of the nature of the data (parametric versus non-parametric).

There are inconsistencies in the application of these tests. The distinction between two-tailed and one-tailed tests is not clearly addressed, which may lead to potential misinterpretations of results. Two-tailed tests identify any difference between groups regardless of direction (increase or decrease), while one-tailed tests specifically determine if a treatment is better or worse.

On the current version of the manuscript, with the application of the two-tailed tests, the authors should correct all sentences mentioning statistical significance, as the following example:

"Bacteria-fed sand flies showed a reduction in the total number of parasites per midgut, 7-, and 11-days post-infection with statistical significance (versus controls; $p \leq 0.0005$; Fig. 2b)" should be corrected to: "Bacteria-fed sand flies showed significant difference in the total number of parasites per midgut, 7-, and 11-days post-infection (versus controls; $p \leq 0.0005$; Fig. 2b)"

However, I highly recommend replacing these tests by their one-tailed versions, so that clear conclusions about the direction of the experiments (increase/decrease) can be drawn.

As additionally requested by Reviewer #5, the confidence intervals are now clearly visible in the plots.

2) Mathematical model (R0)

The new section in the Methods about the R0 models is a clear improvement as suggested by Reviewer #5, as details about this approach were missing on a previous version of the manuscript. The text now adequately describes how the model was applied, its assumptions and limitations. Upon reviewing this section, I understood the model and I agree that its results support the conclusions about the impact of the bacterium treatment on leishmaniasis transmission.

The legend of Figure 6e can be improved for clarity. I would suggest clearly mentioning the difference between the top and bottom panels (earlier in the legend text), and that the diagonal lines and colour shades represent the R0 values.

An additional suggestion, though not mandatory, is to incorporate new points at the ends of the arrows. This would clearly illustrate to the reader the updated values of the mortality and transmission rates associated with the R0 values in the control scenarios. Please refer to the section below for a suggested code snippet for this addition.

3) Code reproducibility

I was also able to fully reproduce the provided R code for the application of R0 models and the respective plot design (Fig 6e), except for a minor typo at line 80 of the code (capital X on the `l_x_bacteria` object):

```
#p_x_bacteria= l_X_bacteria[-1]/l_x_bacteria[-length(l_x_bacteria)]  
p_x_bacteria= l_x_bacteria[-1]/l_x_bacteria[-length(l_x_bacteria)]
```

The R code is well commented, and the sources of the hard-coded values are clearly explained. I suggest uploading it to a code repository (GitHub, GitLab) to facilitate reproducibility and skipping the step of manually copy/pasting it from the supplementary document file.

Like Reviewer #5, I lack the expertise in bioinformatics to assess the reproducibility of the metagenomic analysis code on the GitHub repository.

If the authors would like to add the points to Fig 6e as suggested above, I suggest adding the following in the end of each plot code:

```
# top panel  
points(c((mu_x_b_14),(mu_x_b_14),(mu_x_b_14))~c(1.20*(mu_x_c_14),1.271*(mu_x_c_14),1.90*  
(mu_x_c_14)),pch=19,cex=0.75)  
# bottom panel  
points(c((mu_x_b_14),(mu_x_b_14),(mu_x_b_14))~c(1.20*(mu_x_c_14)*beta_change,1.271*  
(mu_x_c_14)*beta_change,1.90*(mu_x_c_14)*beta_change),pch=19,cex=0.75)
```

And then slightly move the arrow tips so they don't overlap with the new points:

```
# top panel
arrows(x0=1.20*(mu_x_c_14),y0=(mu_x_c_14),x1=1.20*(mu_x_c_14),y1=(mu_x_b_14-0.01),lwd=2,col="white",length =
0.075)
arrows(x0=1.271*(mu_x_c_14),y0=(mu_x_c_14),x1=1.271*(mu_x_c_14),y1=(mu_x_b_14-0.01),lwd=2,col="white",length =
0.075)
arrows(x0=1.90*(mu_x_c_14),y0=(mu_x_c_14),x1=1.90*(mu_x_c_14),y1=(mu_x_b_14-0.01),lwd=2,col="white",length =
0.075)

# bottom panel
arrows(x0=1.20*(mu_x_c_14),y0=(mu_x_c_14),x1=1.20*(mu_x_c_14)*beta_change+0.002,y1=
(mu_x_b_14),lwd=2,col="white",length = 0.075)
arrows(x0=1.271*(mu_x_c_14),y0=(mu_x_c_14),x1=1.271*(mu_x_c_14)*beta_change+0.002,y1=
(mu_x_b_14),lwd=2,col="white",length = 0.075)
arrows(x0=1.90*(mu_x_c_14),y0=(mu_x_c_14),x1=1.90*(mu_x_c_14)*beta_change+0.002,y1=
(mu_x_b_14),lwd=2,col="white",length = 0.075)
```

(Remarks on code availability)

Version 2:

Reviewer comments:

Reviewer #7

(Remarks to the Author)

I thank the authors for addressing all my previous comments on the statistical analysis. After reviewing the revised manuscript, I recommend its acceptance for publication. Congratulations on your work.

(Remarks on code availability)

Point-by-point answers to the Reviewer's comments

Reviewer #1

Key results

*The authors found that treatment of female *P. duboscqi* with *Delftia tsuruhatensis* caused a reduction in the number of *L. major* promastigotes in their guts. The effect was induced only by treatment with live bacteria; the culture supernatant did not affect the infection and the specificity of the effect was confirmed by experiments with other bacterial species (*E. coli*, *O. massiliensis*) and dead *D. tsuruhatensis*, where *Leishmania* infection was unaffected. *Delftia tsuruhatensis* did not alter the alpha diversity of the bacterial community in the gut of *P. duboscqi*, but did affect the relative abundance of different bacterial genera. Specifically, it induced a dominance of the genus *Serratia*, harming *Leishmania* infection. I appreciate that the authors tested the effect of *D. tsuruhatensis* treatment not only before *Leishmania* infection but also in females infected first by *Leishmania*. And valuably, they included experiments where female sand flies were infected naturally by feeding on the host: the difference between the control and treated groups was confirmed even under this arrangement. In terms of potential use in the wild, it is remarkable that significantly higher mortality of sand flies compared to controls was observed only in infected blood fed females. The reduction of *Leishmania* transmission to the host in the treated group of sand flies is impressive; moreover, infected mice were often asymptomatic. Finally, the authors also used a mathematical model to verify that colonization of sand flies by *D. tsuruhatensis* has the potential to disrupt transmission in endemic sites.*

Validity and analytical approach

The vast majority of the results are valid, well analysed and interpreted. Data are mostly presented in graphs; sample size and the statistical tests used are always indicated.

Response: First, we want to thank Reviewer #1 for the time spent reading and evaluating this manuscript. The authors appreciate the thorough revision and the positive evaluation. We are also thankful for the relevant comments that helped us to considerably improve our manuscript. We considered all of the issues raised by the Reviewer and revised the manuscript, when applicable (yellow highlighted text); please check below our point-by-point responses to your specific comments.

The only missing test of statistical significance concerns data summarized in the graphs of Fig. 4B, Fig. 5B (difference between the control and treatment groups in the prevalence of infected flies and mice that developed lesions, respectively).

Response: Thank you for your feedback. The reviewer is correct; we did not conduct a statistical analysis within the framework of the two datasets mentioned, primarily because we used this presentation as a graphical aid. In the revised version of the manuscript, we have included additional analyses related to these data in Figures 4B and 5B (now 5b and 6b), as well as the data in Figure 5e, and have referenced them in the main text. Specifically, we calculated:

- Fig. 5b: The relative risk of a sand fly being infected with *Leishmania* parasites 11 days post-infection in the context of pickup experiments with bacteria-fed versus control sand flies – 0.6491 (95% CI of 0.4277 to 0.9820; $p=0.0482$; Fischer's exact test). This means that a bacteria-fed sand fly has around a 1.54 times lower chance of sustaining a *Leishmania* infection than a control sand fly.

- Fig. 5e: The relative risk of an infected sand fly dying 6 days after taking a second, non-infected bloodmeal comparing bacteria-fed versus control sand flies – 7.216 (95% CI of 4.224 to 12.78; $p < 0.0001$; Fischer's exact test). This means that a *Leishmania*-infected bacteria-fed sand fly has around a 7.26 times higher chance of dying than a control sand fly after a second blood meal.
- Fig. 6b: The relative risk of a mouse developing a visible cutaneous leishmaniasis lesion after being bitten by *Leishmania*-infected bacteria-fed versus control sand flies – 0.2361 (95% CI of 0.08004 to 0.6084; $p = 0.002$; Fischer's exact test). This means that an animal bitten by *Leishmania*-infected bacteria-fed sand flies has around a 4.235 lower chance of developing a cutaneous leishmaniasis lesion than an animal bitten by *Leishmania*-infected control sand flies.

Methodology

The authors used adequate methods that are sufficiently described in the supplementary materials. I just have a comment on the evaluation of sand fly infections: the effect of D. tsuruhatensis on Leishmania infection is presented in terms of the number of promastigotes in guts. I am surprised that there is no assessment of the prevalence and localization of infections; I only found the prevalence of infections initiated by feeding on the host. Could authors provide comparison of percentage of infected females in treated and control groups for artificial membrane feedings?

Response: We would like to thank Reviewer #1 for agreeing that, overall, the methods we used were adequate, and are well described. As per the issue raised regarding the prevalence of infection, we have now included these results for all groups in the updated figures. Of note, we chose an artificial infection setting that pushes the system, in a way that any visible phenotype would be definitely relevant – our infection protocol ensures that often 100% of blood-fed sand flies get heavily infected, homogeneously, and that the whole population of sand flies develop mature infections with high proportion and number of metacyclics that will ensure disease transmission; this is likely not what we will be observed in the field. Thus, we still only highlight in the manuscript text the impact of TC1 bacteria treatment on the prevalence of sand fly infection in the context of the pickup model (a system that mimics natural infection; data in Figure 5b), which is undeniable.

Regarding parasite localization, it is important to know whether promastigotes colonize the stomodeal valve in mature infection. I like to ask authors to show whether the proportion of females with colonized stomodeal valve differed between the control and treatment groups. However, if the authors do not have these data, I do not insist on completing the additional experiments because transmission to mice was demonstrated.

Response: Thank you for your remark. Indeed, we did not initially focus on parasite localization. However, recognizing its importance, we conducted a new set of independent experiments, which are now included in the revised manuscript. While no major differences were observed macroscopically, we identified a significant difference in the diameter of the anterior midgut of control versus bacteria-fed, *Leishmania*-infected sand flies. These findings align with the data on feeding success in the context of transmission experiments, which indirectly suggested that sand flies colonized with the bacteria appeared to be "less blocked." This new data is now included in the revised Figure 2 (Fig. 2e) and is described in the text as follows:

“...when we compared the diameter of the anterior midgut of *D. tsuruhatensis*-fed versus control sand flies, we observed that the former showed smaller diameters, with statistical significance ($p=0.0247$; Fig. 2e).”

Significance

*This work represents original research with stimulating implications for leishmaniasis control. It shows that *Delftia tsuruhatensis*, which has previously been used successfully in *Anopheles* mosquitoes to reduce their ability to transmit *Plasmodium*, is also effective in reducing *Leishmania* transmission by sand flies under laboratory conditions. We can only wish that this efficacy is confirmed in the field.*

Response: We are grateful for this evaluation. We too share the hope the modelling predictions become a reality once we take this intervention to the field and beyond.

However, it is also important to point out the safety risks associated with this control application, which I recommend to include in the discussion.

Response: Thank you for this comment. We now extend the discussion on the safety risks of such an intervention in the discussion section of our revised manuscript. Please check our response to your last point for further details.

Clarity and context

The text is very clearly written, and the chapters with results are logically linked. The Introduction and Discussion are not too lengthy, yet they cover important facts and discuss relevant previous results by other authors. However, it would be helpful to add a few points (specified below) to the discussion. I have some minor comments in the following paragraph:

Response: Again, we thank the Reviewer for the positive comments! As per the constructive feedback, we have considered all of the issues raised. Please check specific responses below.

- Abstract should contain no references.

Response: We apologize for this oversight. Now, the Abstract is unreferenced, as per the journal guidelines.

- Line 56 – reference number 8 is not optimal, an older publication like Maroli 2012 (doi: 10.1111/j.1365-2915.2012.01034.x) should be cited.

Response: We used the recommended reference instead. Please check new reference #8.

- I propose to integrate the results of the chapter *Specificity of the Delftia tsuruhatensis effect on Leishmania infection (Fig. 1 and Fig. S1)* into a single figure. The scheme in Figure S1 could be simplified, e.g., by omitting the image of the dissected sand fly.

Response: We agree with the reviewer. Therefore, we have updated the figures as follows:

- Fig. 1 now only refers to colonization.
- Fig. 2 is now a combination of panels C and D of the original Fig. 1, as well as both panels of the original fig. S1. Additionally, the data referring to the diameter of the stomodeal valve in control and bacteria-fed infected sand flies was also included in this figure.
- fig. S1 was deleted, and the remaining supplementary Figures were re-numbered.

- Lines 112-113: *If there is a single species of the genus Serratia (S. ureilytica), the abbreviation Serratia sp. should be used. The abbreviation Serratia spp. would correspond to a situation where there are multiple species in the sample.*

Response: Thank you for the note. We maintained "spp." in the first mention of *Serratia* for accuracy, as we are referring to the genus, which includes several species. However, we corrected the second mention to "sp." since we are indeed referring to a single species, as the reviewer highlighted. Notably, we kept the usage of "spp." after *Delftia*, as we cannot ensure we are only looking at a single species (e.g., *Delftia* – but not the TC1 strain - is present in the midguts of control sand flies – Fig. S3a). Therefore, the sentences now read:

“Conversely, in *D. tsuruhatensis*-fed sand flies a clear prevalence of a single bacterial genus, *Serratia* spp., was evident at all time-points (Fig. 3e). Of note, while the relative abundance of *Serratia* sp. (identified as *Serratia ureilytica* via shotgun sequencing; accession number: SAMN40248728) was higher than that of *Delftia* spp. (fig. S3a), the estimated absolute abundance of both *Serratia* sp. and *Delftia* spp. consistently increased in a similar order of magnitude in bacteria-fed versus control sand flies (Fig. 3f).”

- Lines 263-264: *“To our knowledge, no vector control approach aiming to promote refractoriness was deemed effective in the field for both malaria and leishmaniasis.” Are the authors referring to the same product that would be universally effective against both Plasmodium in mosquitoes and Leishmania in sand flies? It should be worded more clearly.*

Response: Thank you for the question. The manuscript was updated for clarity. The text now reads:

“To our knowledge, to date, no single vector-based control approach aiming to promote refractoriness was deemed effective in the field for the simultaneous control of malaria and leishmaniasis (targeting both *Leishmania* spp. parasites within sand flies, and *Plasmodium* spp. parasites within mosquitoes, respectively).”

- In Discussion, authors explain the observed phenotype by dysbiosis of sand fly guts. What is the authors' opinion on the possible influence of the sand fly immune response on the observed effect? For example, it has been described that defensins are upregulated in *Leishmania* infected *P. papatasi* (10.3390/microorganisms9112307).

Response: Thank you for the question. We thought about anti-bacteria immunity as a possible explanation for our phenotype. However, the results we obtained with dead *D. tsuruhatensis* (lost of phenotype), suggest that a strong immune response to *Delftia* PAMPs is likely not occurring in our context. Do note that *Delftia* strains are reported as very efficient competitors under different conditions [1-3]. Therefore, what we think is most likely happening is a “re-organization” of the sand fly gut microbiome in response to *Delftia* excreted-secreted anti-bacterial effectors; particularly, this is leading to the decrease in the relative abundance of several bacterial genera and simultaneously, the increase in the relative abundance of few genera, including striking changes with regards to *Serratia urealytica*. This said, we cannot exclude there is a combination between *Delftia*-direct effects and sand fly immunity-related effects behind the gut dysbiosis phenotype reported in our paper. For instance, we cannot exclude an immune response induced, not by *Delftia*, but by *Serratia* PAMPs. As per the contribution of *Leishmania* parasites for the induction of sand fly immunity to an extent that leads to dysbiosis, we do not think that is the case here; both control and bacteria-fed groups were infected, and the differences between them are dramatic. Of note, bacteria-fed sand flies show lower infection rates than control ones. While such a discussion is interesting, because our data seems to exclude a major impact of direct immune responses to both *D. tsuruhatensis* bacteria and *Leishmania* parasites to the phenotype, we decided not to include it in the revised version of the manuscript. This said, we did mention briefly a potential role of sand fly immunity in our phenotype, because it is something that we cannot totally exclude, as follows:

“Conversely, the expansion of *S. ureilytica* after *D. tsuruhatensis* colonization may be detrimental to the development of *Leishmania* within sand flies (a role of sand fly immunity cannot be excluded in this context).”

- In Discussion, I would appreciate an expanded passage on the potential risk of introducing the model in the field. Since the composition of the microbiome of wild sand flies is variable and differs from that of a laboratory colony, it is difficult to predict what changes in the composition of the bacterial community will occur after *Delftia tsuruhatensis* treatment. A bacterial species may be selected that will support the development of *Leishmania* (e.g. 10.1128/mBio.01121-16, 10.1111/cmi.12755) or species pathogenic to humans. For example, a new opportunistic pathogen, *Serratia marcensens*, exhibiting resistance to many antibiotics (10.3390/antibiotics12020367, 10.7717/peerj.14399) has been described from wild sand flies (10.1155/2020/5458063, 10.1371/journal.pone.0050259).

Response: Thank you for the comments. In the new version of the manuscript the above listed limitations were further discussed. With respect to the unknown effect on wild caught sand flies, we simply highlighted the need for field studies, as follows:

“Therefore, the dysbiosis-related effect on *Leishmania* within sand flies observed here, should also apply to wild insects. This said, studies with wild-caught sand fly populations are needed to understand if the expected induction of dysbiosis will still promote refractoriness to *Leishmania* spp. infection.”

As per the second point, we agree that the induction of sand fly dysbiosis may lead to the selection of human pathogenic bacteria. We would use the mentioned references to highlight the potential of transmission of multi-drug resistant strains, but the papers on the characterization of the sand fly microbiota do not mention the status of susceptibility of the *Serratia* spp. strains (10.1155/2020/5458063, 10.1371/journal.pone.0050259), and thus we cannot extrapolate these are the same as the one reported in the studies on the susceptibility of *Serratia* isolates (10.3390/antibiotics12020367, 10.7717/peerj.14399). Instead, we mentioned the simple possibility of the occurrence of secondary infection without forgetting an important consideration. These possible secondary co-infections are probably initiated after the co-egestion of bacteria and parasites in the context of heavily “clogged midguts” (via regurgitation). Sand flies with intact stomodeal valves (particularly the non-infected ones) are not expected to “egest” (at least in high amounts) gut bacteria. Sand flies with smaller plugs are also expected to egest less bacteria (they are expected to feed easier). Since the introduction of *D. tsuruhatensis* leads to lighter infections (and sand flies that feed easier – Fig. 6a), we may actually be promoting protection from secondary infection (in theory less bacteria egested as compared with a control sand fly). We need to consider that secondary infection is a possibility in the context of a bite by any wild heavily infected sand fly. We also need to consider that the secondary infections reported may be posterior to parasite inoculation, and just a consequence of the disease (VL and immune-suppression; CL and the breakage of the skin layer). Based on these considerations, our discussion was updated as follows:

“Importantly, *Delftia* spp. are environmental bacteria, widely distributed in nature³³, and are part of the microbiota of different human-disease vectors, including sand flies, mosquitoes, and ticks^{11,34}. Therefore, the use of *D. tsuruhatensis* TC1 for the vector-based control of leishmaniasis is not expected to disturb natural ecosystems. However, the changes in the microbiota of wild hematophagous insects may be associated with risks, particularly when we consider the possibility of indirect pathogenic effects. We know that, together with *Leishmania* parasites, sand flies regurgitate a complex inoculum into the skin¹, including sand fly midgut bacteria³⁵. This has the potential to lead to *Leishmania* spp.-bacteria co-infection scenarios that are sporadically reported³⁶. Of note, most of these scenarios are probably just a consequence of the progression of leishmaniasis and not of vector bites-derived inoculation; visceral leishmaniasis is known to induce immune suppression³⁷ and cutaneous leishmaniasis may lead to the compromise of the skin barrier function³⁸, both risk factors for the establishment of opportunistic pathogens. Of relevance, the inoculation of gut bacteria via sand fly bites is dependent on the infection status; the higher the infection, the greater the probability of regurgitation to allow blood-feeding^{12,13}. Therefore, the fact that *D. tsuruhatensis* colonization leads to a decrease in the sand fly *Leishmania*-infection burden, may suggest these sand flies are egesting lower levels of gut bacteria into the host skin, making it less probable for these insects to transmit any resilient bacterial agent that becomes prevalent in a sand fly gut dysbiosis context. Of note, excluding CL lesions, no visible clinical manifestations were observed in our animals, including those exposed to the bites of *D. tsuruhatensis*-colonized sand flies. Adding to this, no *Delftia* secondary infection in the context of leishmaniasis was reported to date, whereas *Serratia* spp.-*Leishmania* spp. co-infection was reported only once³⁶. This said, in depth studies on the potential co-transmission of potentially pathogenic bacterial strains with *Leishmania* spp. parasites in the context *D. tsuruhatensis* TC1 colonization need to be carried out to support the development of any mitigation strategies, if needed.”

Reviewer #2

After carefully reading the manuscript from Cecilio et al., the study appears to be scientifically well designed, and the data generated looks well analyzed and very interesting. The discussion around those data is well elaborated, although one of my key points of suggestion is about the latter part of the discussion.

Response: First, we want to thank Reviewer #2 for the time spent reading and evaluating this manuscript. The authors are very grateful for the positive evaluation of our work. We are also thankful for the relevant comments that helped us to considerably improve our manuscript. We considered all the comments and revised the manuscript (yellow highlighted text), when applicable; please check below our point-by-point responses to your comments.

My minor comments are below:

*Line 1: In the title, I suggest adding what TCI symbiotic bacterium is or something that could give more details instead of just saying *Delftia tsuruhatensis* TCI.*

Response: Thank you for the suggestion. We understand the reviewer's point. Indeed, *Delftia tsuruhatensis* TCI was identified as a symbiont of *Anopheles* mosquitoes. The fact that this bacterium does not impact the fitness of non-infected sand flies may suggest that it is also a symbiont in the context of these insects. This said, we cannot exclude that it is not the case. Therefore, having your suggestion in mind, we decided to update the title, as follows, for the sake of accuracy:

“*Leishmania* sand fly-transmission is disrupted by *Delftia tsuruhatensis* TCI **bacteria”.**

Lines 269-272: The use of sugar baits adapted for targeted delivery of TCI-based strategies needs to be discussed more. A recent study with malaria mosquitoes showed that these sugar bait strategies did not work very well with ATSB in Zambia: Wagman et al. 2024, "Entomological effects of attractive targeted sugar bait station deployment in Western Zambia: vector surveillance findings from a two-arm cluster randomized phase III trial." Considering that most malaria-endemic countries in Africa have humid tropical weather with a lot of nectar naturally available for mosquitoes, this makes sugar baits less competitive than natural sugar sources. This should be discussed, focusing on the fact that sugar baits could work in environments like Sahelian malaria-endemic countries such as Mali, where attractive sugar baits seem to work. In other contexts, authors could think about other strategies, like larval breeding seeding with TCI or releasing some fraction of TCI-infected sandflies to be disseminated. This delivery component is crucial for the deployment of TCI-based strategies for sandfly control and even for mosquitoes as well.

Response: Thank you for the relevant comments. Indeed, the deployment strategy needs to be carefully considered for the success of the intervention we are proposing. With regards to targeting the larval stage, we think it won't be feasible for sand flies. While for mosquitoes we can introduce the bacteria in larvae-containing water sources and expect easy bacteria dispersion and consequently a good colonization coverage, the same is not true with respect to sand fly larvae. Sand flies do not have an aquatic stage; larvae develop on organic matter rich soils[4]. Therefore, identifying areas in the wild that are rich in sand fly larvae remains challenging. Do note that we were able to verify in the laboratory setting that *Delftia tsuruhatensis* TCI passes transtadially from immature forms to sand fly adults. However, to achieve a good population coverage, we had to give the bacteria to larvae on a weekly basis, during the full larval development period (versus a single deployment in adult sand flies).

As per the targeting of adults, we do understand the potential drawback highlighted in the context of settings with abundant natural sugar sources available. Our idea is to devise baits containing strong attractants (there are even odorant molecules that seem to be attractant to both sand flies and mosquitoes, e.g, 1-octen-3-ol [5, 6]). We hope to test in the future different odorant molecules under semi-field conditions to select the better option (either a single compound or a combination of compounds). We also plan to devise some experiments including alternative natural sugar sources to try to understand their impact on the final colonization outcome.

The above notions were included in the revised Discussion section of our manuscript, as follows:

“...Considering that malaria and leishmaniasis are co-endemic in more than 50 countries worldwide^{42,43}, our data is of great relevance. Our results regarding sand flies, together with those recently reported with mosquitoes⁵, support the development of *Delftia tsuruhatensis* TC1-based interventions for the concurrent control of leishmaniasis and malaria in the field. Of note, while for mosquitoes, the targeting of the larval stage is an option, for sand flies such an approach would be quite challenging. Unlike mosquitoes, sand flies do not have an aquatic larval stage; they lay their eggs in organic matter-rich environments, which are difficult to locate and mostly unknown¹. Therefore, the reduced potential for bacterial dispersion in soil (versus water) makes deploying *Delftia tsuruhatensis* TC1 in such non-aquatic sites unfeasible. With this in mind, we envision, instead, the use of sugar baits adapted for targeted delivery, in places where the insects are expected to occur (e.g. breeding sites). Importantly, such sugar baits will not only contain the bacterial suspension for delivery, but also specific odorant-attracting molecules to make them more competitive than natural sugar sources (particularly in places where such sources are abundant), and prevent potential sub-optimal deployment, as reported elsewhere in a different context⁴⁴. Of note, several molecules were reported to attract mosquitoes and sand flies, including some that have the potential to attract both insect species simultaneously (e.g. 1-octen-3-ol)^{45,46}. The development of these sugar baits is expected to be done in the near future, for use in the context of semi-field studies that will allow us the extrapolation of our results to the field context.”

Reviewer #4

*This study demonstrates that *Delftia tsuruhatensis* TC1 can colonize the midgut of sandflies without affecting their growth and development. Meanwhile, *Delftia tsuruhatensis* TC1 can specifically affect the growth and development of *Leishmania* and reduce their transmission ability, effectively reducing the probability of *Leishmania* outbreaks. The research approach is novel, the results are practical and feasible, and can be accepted after revisions.*

Response: We would like to thank Reviewer #4 for the time spent reading and evaluating this manuscript, and for the positive evaluation of our work. We are also thankful for the relevant comments that helped us to considerably improve our manuscript. We considered all the comments and revised the manuscript (yellow highlighted text), when applicable; please check below our point-by-point responses to your comments.

Here are my suggestions:

*Q1: Why choose *Delftia tsuruhatensis* TC1? How effective is *Delftia tsuruhatensis* TC1 in killing *Leishmania* compared to other bacterial strains or drugs? Comparison is suggested to be made.*

Response: Thank you for the question. The rationale here was quite simple. Since we isolated an environmental bacterial strain able to colonize the midgut of *Anopheles* spp. mosquitoes and impact the development of *Plasmodium* spp. (with a consequent negative impact on the transmission of malaria), we decided to try to understand whether the same bacteria could also be used for the control of *Leishmania* parasites within sand flies. Do note that contrary to *Plasmodium* spp., *Leishmania* spp. parasites spend their entire life in the insect midgut[4, 7], and thus the bacteria-parasite contact is expected to be constant, which may be an advantage. Importantly, as far as we are concerned, no sand fly-based microbiological intervention is available for the control of leishmaniasis (as opposed, e.g. to the use of *Wolbachia* in mosquitoes). Therefore, no good bacterial parallel could be used as the “gold standard” in the context of a comparison of efficacy. Additionally, the “drugs” used for vector control purposes in the context of leishmaniasis, are insecticides, whose mechanism of action is the indiscriminate killing of sand flies, rather than the specific targeting of *Leishmania* infection within the vector. Therefore, also in this context, no good drug versus bacteria parallel is available thinking on a comparative study. Based on the above, we clarified our rationale in the manuscript, as follows:

“Recently, we identified and characterized a *Delftia tsuruhatensis* strain that promotes the refractoriness of *Anopheles* spp. mosquitoes to *Plasmodium* spp. infection⁵. The phenotype was attributed to the specific targeting of *Plasmodium* ookinetes by harmaline, a beta-carboline alkaloid derivative produced by *Delftia tsuruhatensis* TC1⁵. Interestingly, beta-carboline alkaloids have shown some anti-*Leishmania* activity⁶. This, coupled with the fact that, contrary to the development of *Plasmodium* parasites in mosquitoes⁷, the development of *Leishmania* parasites is restricted to the sand fly midgut¹ (and thus are expected to be in constant contact with this bacterial strain and its byproducts), made us hypothesize that *D. tsuruathensis* TC1 would also be a good vector refractoriness-promoting agent for the control of leishmaniasis. In this exploratory study, addressing this hypothesis, we investigated whether *D. tsuruathensis* TC1 can be used to impact the transmission of *Leishmania* parasites by sand flies.”

Q2: How did the author separate *Delftia tsuruhatensis* TC1? Suggest providing detailed supplements in the part of methods and material.

Response: Thank you for the question. All of these details were reported in our previous publication (Huang, W. *et al.* *Delftia tsuruhatensis* TC1 symbiont suppresses malaria transmission by anopheline mosquitoes. *Science* **381**, 533-540, doi:10.1126/science.adf8141; 2023). Isolation was performed as follows:

“Isolation and identification of *D. tsuruhatensis* TC1 as an inhibitor of *Plasmodium falciparum* An *An. stephensi* colony from GSK’s insectary at Global Health Medicines R&D site in Tres Cantos, Spain, gradually lost susceptibility to *P. falciparum*, but mosquito survival and reproductive ability were not compromised. To establish factors that could affect the loss of susceptibility to *P. falciparum*, we explored the microflora present in different mosquito tissues and breeding environments. Homogenized tissues from pupae, larvae, adult female mosquito midguts and water from larval breeding pans were transferred to sterile LB broth and incubated at 27 °C or at room temperature on a rotary shaker. When turbidity was observed, samples were h plated on solid LB agar to obtain single colonies. Colonies were selected based on distinct morphology, transferred to fresh liquid media and subsequently identified by biochemical characterization using API strips (Biomereux) and by 16sRNA typing. A bacterium identified as *Delftia tsuruhatensis* was found to be predominantly present in all screened samples and the strain was designated as Tres Cantos 1 or TC1.”

Do note that we always directed the readers to this study when we talked about the bacteria in the Methods section of our paper (“Our collaborators isolated a *Plasmodium* refractoriness-promoting bacterium from the midguts of these mosquitoes identified as *Delftia tsuruhatensis* and the strain was designated as Tres Cantos 1 or TC1⁵.”). This said, we agree with the Reviewer that, to provide this information in our Materials and Methods section may be important. Consequently, we updated the “**Bacteria**” sub-section of our Materials and Methods, accordingly, as follows:

“An *An. stephensi* colony from GSK’s insectary at Global Health Medicines R&D site in Tres Cantos, Spain, gradually lost susceptibility to *P. falciparum*, but mosquito survival and reproductive ability were not compromised. Our collaborators isolated a *Plasmodium* refractoriness-promoting bacterium from the midguts of these mosquitoes⁵. Briefly, homogenized tissues from pupae, larvae, adult female mosquito midguts and water from larval breeding pans were transferred to sterile LB broth and incubated at 27 °C or at room temperature on a rotary shaker. When turbidity was observed, samples were plated on solid LB agar to obtain single colonies. Colonies were selected based on distinct morphology, transferred to fresh liquid media and subsequently identified both via biochemical characterization using API strips (Biomereux) and via 16sRNA typing. A bacterium identified as *Delftia tsuruhatensis* was found to be predominantly present in all screened samples and the strain was designated as *Delftia tsuruhatensis* Tres Cantos 1 or TC1. A GFP-expressing *Delftia tsuruhatensis* was further generated and used for imaging purposes.”

Q3: What is the concentration of *Delftia tsuruhatensis* TC1 in the study? Is the disruption of *Leishmania* transmission by *Delftia tsuruhatensis* TC1 concentration-dependent?

Response: Thank you for the question. We maintained the concentration of *D. tsuruhatensis* TC1 constant at the delivery stage (1×10^8 CFUs/ml of sugar). The exact amount taken by the sand flies is unclear and is expected to be heterogeneous once it will depend on how much volume sand flies take per sugar-meal, as well as on how many sugar meals sand flies take during the overnight period the bacterial solution is available *ad libitum*. We did observe a dose-dependence tendency (check **Figure R1** below). Now, we include this data on the revised version of the paper as the new fig. S1.

Figure R1. Effect of the dose of *Delftia tsuruhatensis* on the *Leishmania* infection phenotype in sand flies. Sand flies were allowed to take a sugar meal (5% sterile sucrose solution), alone (control), or containing different concentrations of *Delftia tsuruhatensis* (numbers in the x axis/ml), and then infected with *L. major* parasites via artificial membrane feeding. Midguts were dissected 11 days post-bloodmeal and the infection burden was quantified microscopically. The total number of parasites per midgut of control (black dots) and bacteria-fed (red dots) sand flies is represented. Each dot represents a single midgut. The group median and 95% CI are also shown. Statistical significance was determined using the Kruskal-Wallis test followed by a post-hoc analysis with the Dunn’s test, and is highlighted: ns – not significant; and ** $p \leq 0.01$. Of note one exact p value is shown due to its “borderline significance”. All results were obtained in, at least, two independent experiments.

Q4: Why choose *Escherichia coli* for specific research? I think using only one experiment with *Escherichia coli* cannot draw a conclusion on whether it is specific or not, and it does not have universal

significance. More bacterial strains experiments are expected to demonstrate the specificity of Delftia tsuruhatensis TC1. And how to determine if Escherichia coli has successfully colonized?

Response: Thank you for the question. We used *E. coli* as the prototypical Gram-negative bacteria (*D. tsuruhatensis* TC1 is a Gram negative); therefore, if the phenotype was a consequence of a response to e.g. Gram-negative bacterial PAMPs, the use of *E. coli* would be a good parallel. Of note, *E. coli* was reported to be present on the microbiota of sand flies [8], and therefore, it is not unreasonable to consider that it may colonize the midgut of sand flies. This said, we were aware of the highlighted limitation. While we are sure the delivery of the same dose of *E. coli* would provide a similar initial immune stimulus, we are not sure that *E. coli* colonized the midgut of our sand flies and then we could not undoubtedly extrapolate to the context of an “active infection” (e.g. we can be using a strain not adapted to sand flies); This was the reason why we used an additional bacterial agent, directly isolated from the microbiota of *P. duboscqi* sand flies from our colony – *O. massiliensis*; this one is expected to easily colonize the midgut of our sand flies. The absence of a phenotype in both contexts, together with the data obtained with dead *D. tsuruhatensis* TC1 is, in our opinion enough for us to conclude that the effect is *D. tsuruhatensis* TC1-specific, and neither a consequence of a general bacterial infection, nor a consequence of immune responses directed to bacterial PAMPs. This said, we performed a couple of more experiments with one additional bacterial strain isolated from the microbiota of *P. duboscqi* sand flies from our colony (*Enterococcus faecalis*), that again, did not impact the development of *Leishmania* parasites within the sand fly midgut. We believe that these results, now included in revised Fig. 3 (new panel c) and discussed in the revised manuscript, strengthen our claim that the phenotype is a result of colonization with live *D. tsuruhatensis* bacteria, and neither a consequence of a “general bacterial infection”, nor a consequence of a general response to bacterial PAMPs.

Q5: How to determine if Delftia tsuruhatensis TC1 has died in specificity experiments? What processing method led to the death of Delftia tsuruhatensis TC1? Is the supernatant of dead Delftia tsuruhatensis TC1 used in the experiment or bacterial cells?

Response: Thank you for the questions. As initially stated in the Materials and Methods section of our manuscript, we killed the bacteria via heating at 95 °C for 10 minutes (“For experiments with killed bacteria, the bacterial suspension was incubated at 95 °C for 10 minutes prior to dilution in sucrose solution.”- Bacterial growth and preparation sub-section). Do note that *Delftia* spp. bacteria are not expected to sporulate [3]; therefore, highly resistant structures are not expected to be found in a bacterial suspension of *D. tsuruhatensis* TC1. This said, to objectively answer to your question, we did a process of heat-killing and then plated the suspension; expectedly (**Figure R2** below), we did not see any bacterial growth after a 3-day incubation period compared to control (we normally see bacterial growth after an overnight incubation). This Figure was now included as an additional supplementary Figure in our paper (new fig. S5). It is also important to mention that, when we mean killed bacteria, we mean the actual bacteria (or bacteria whole lysates) and not the resulting supernatant. Overall, briefly, first we separate the amount of bacteria to deliver, and only then proceeded with the heat-inactivation, to account for possible changes in the turbidimetry of the solution after this process and the consequent possible mis-estimation of the bacterial concentration. Afterwards, we just dilute the resulting suspension (no prior centrifugation step) in the respective volume of sucrose and use this suspension to soak the cottons for delivery to sandflies.

Figure R2. Heat kills *Delftia tsuruhatensis* bacteria. *D. tsuruhatensis* TC1 were grown overnight in LB broth at 30 °C under agitation (200 RPM) and washed with sterile PBS via sequential centrifugation steps (3500 RPM, 10 minutes 4 °C). Two aliquots of 5x10⁷ colony forming units (CFUs) were then suspended in 70 µl of sterile PBS. One aliquot was then kept in the lab bench (control), while the other was

incubated at 95 °C for 10 minutes. Soon after, each suspension was plated via spreading into an individual blood agar plate, and plates were incubated at 30 °C for 3 days. Images of both plates after the incubation period are shown.

Q6: The supernatant and bacterial derivatives do not affect the growth of Leishmania parasites, indicating that live Delftia tsuruhatensis TC1 are the ones that are affected. So, how does live Delftia tsuruhatensis TC1 affect the growth of Leishmania?

Response: Thank you for the comments. In our experimental context, harmaline did not affect the growth of *Leishmania* parasites both *in vitro* (fig. S2d) and *in vivo* (fig. S2e). However, the supernatant **did** affect the growth of *Leishmania* parasites *in vitro* (figs. S2b and c). While the same was not true in the *in vivo*, our results suggest that some bacteria excreted-secreted products are, indeed leishmanicidal. There are a few explanations that may support these disparate *in vivo* and *in vitro* results.

The first one has to do with the delivery. While we know the bacteria targets the gut, we cannot ensure the same applies to the supernatant. We are delivering via sugar-meal (except when we use 100% supernatant), known, contrary to the blood-meal, to be routed to the crop and not the midgut [9].

The second (which may be related with the first) has to do with the effective concentration of bacteria excreted-secreted products on the midgut. Live bacteria are expected to constantly secrete the leishmanicidal molecules directly into the midgut (in concentrations that may be much higher than those obtained in the context of culture in rich liquid medium under optimal conditions – the bacteria are expected to compete for space and resources [10], and therefore are under a much greater stress). On the other hand, we are delivering a defined amount of supernatant via sugar-meal. Of note, the amount ingested will depend on the volume of “sugar” ingested and the number of times sand flies take a sugar-meal. Additionally, because of this system, the delivery cannot be considered a “constant delivery”. These facts together with the possibility of the sugar-meal not being routed to the midgut, will ultimately influence the effective concentration of *D. tsuruhatensis* leishmanicidal byproducts in the sand fly midgut. Adding to the above, we cannot exclude the occurrence of some degradation of bacteria-secreted products within the sand fly midgut. Consequently, we can hypothesize that probably, the continuous delivery of such byproducts by live bacteria within the midgut is enough to ensure a concentration above the minimum inhibitory concentration necessary for an effect on *Leishmania* establishment, while the delivery of the supernatant is not.

One small note on harmaline, while our data shows that in our context this molecule alone does not negatively impact *Leishmania major* parasites, we cannot definitely say that this molecule is not responsible for the phenotype we are seeing with the supernatant *in vitro*, or live bacteria *in vivo*. For instance, this molecule may act synergistically with other bacteria excreted-secreted products in the killing of *Leishmania*.

Overall, we think that our phenotype is due to a combination of factors: i) a direct effect of bacteria secreted/excreted products with leishmanicidal properties on *Leishmania* parasites, and ii) and an indirect effect of *D. tsuruhatensis*-induced sand fly gut dysbiosis on *Leishmania* parasites (of note we know that the gut microbiota is important for the establishment of *Leishmania* parasites in sand flies [9, 11] – please check the response to the metagenomics analysis point, below, for more details). Do note that this hypothesis was written in the discussion section of our original manuscript (“Of note, although an inhibition effect of the bacterial supernatant was not observed *in vivo*, our data does not exclude the hypothesis of a direct effect of *D. tsuruhatensis* TC1 on the establishment of *Leishmania* in sand flies. Therefore, we may speculate that our phenotype can both be a consequence of direct, and dysbiosis-

related indirect effects of *D. tsuruhatensis* midgut colonization.”). This said, we updated our discussion, for clarity, as follows:

“Of note, although an inhibition effect of the bacterial supernatant was not observed *in vivo*, our data does not exclude the hypothesis of a direct effect of *D. tsuruhatensis* TC1 on the establishment of *Leishmania* in sand flies (e.g. the anti-*Leishmania* component(s) in the bacterial supernatant may be rapidly degraded within the sand fly midgut, and thus, a constant *in situ* secretion of such component(s) by live bacteria is needed for the successful control of *Leishmania* spp. parasites). Therefore, we may speculate that our phenotype can be a consequence of both, a direct effect of bacteria excreted/secreted products on *Leishmania* spp. parasites, and an indirect effect consequence of *D. tsuruhatensis* TC1-induced sand fly gut dysbiosis.”

Q7: It is best to indicate the number of experimental subjects in each group in each figure. Why is the number of repetitions different for each group in Figure 2D?

Response: We apologize for this oversight. We indicate the “n” for all the datasets either at the Figures or at the Figure legends). As per Fig. 2D (now Fig. 3e) each sample is composed of a pool of 20 sand fly midguts from three independent experiments. We collected extra replicates when the number of sand flies allowed in both treatment conditions.

*Q8: Will the colonization of *Delftia tsuruhatensis* TC1 affect the vitality and growth of sandflies?*

Response: Thank you for the question. We believe the fitness data we presented in the original version of our manuscript answers, at least in part to this question:

“In line with the above, we looked at sand fly survival. Always comparing *D. tsuruhatensis*-fed with control sand flies, we determined the mortality, daily, for sand flies that never took a bloodmeal, as well as for those that took a non-infected or a *Leishmania*-infected bloodmeal. In the absence of blood, mortality was negligible in both *D. tsuruhatensis*-fed and control sand flies (Fig. 5c, left panel). In the presence of non-infected blood, mortality increased for both groups (Fig. 5c, middle panel). Interestingly, in the context of infected blood, we saw a clear separation of the survival curves of control and *D. tsuruhatensis*-fed sand flies; the latter showed a lower survival rate throughout the monitoring period (Fig. 5c, right panel). Overall, these data translated into a significant difference in the survival of, only *Leishmania*-infected *D. tsuruhatensis*-fed versus control sand flies (data from day 11; $p=0.0383$; Fig. 5d). Strikingly, the increase in mortality was even more pronounced in infected sand flies that took a second bloodmeal (25% survival versus 90% in the control group; Fig. 5e). Therefore, the use of *D. tsuruhatensis* TC1 has the potential, not only to decrease the infection burden of sand flies, but also to reduce the number of infected sand flies in nature.”

Altogether, our data suggests that *D. tsuruhatensis* do not affect the fitness (or vitality) of sand flies, unless they are infected. Do note that the proportion of infected flies in the field is low (from 0.7 to 2% according to different studies, although this may be an underestimation [12]); therefore, a huge effect of this decreased fitness of infected sand flies is not expected to lead to dramatic changes in the overall population.

As per the growth, we assume the reviewer means the reproductive output. To address this, we did an experiment considering only the infected sand fly population, since this is the only one whose fitness is affected by *D. tsuruhatensis* colonization, as highlighted above; we are assuming here that the increased mortality might affect the reproductive output (e.g. flies may die before laying their eggs). We counted the viable progeny generated by 150 females either colonized or not by *D. tsuruhatensis* and infected with *Leishmania* parasites, in the context of 3 independent experiments. While we saw a tendency of decrease of the number of F1 emerging adults sired by *Leishmania*-infected bacteria-colonized versus control sand flies, this was not statistically significant (see **Figure R3** below, for Reviewers' eyes only). Again, bear in mind that the proportion of infected flies in the field is low, and thus, such a decrease is not expected to affect the overall "growth of sand flies", which we again are assuming means the replacement of sand fly generations.

Figure R3. Effect of *Delftia tsuruhatensis* colonization on the reproductive output of *Leishmania*-infected sand flies. Sand flies were allowed to take a sugar meal (5% sterile sucrose solution), alone (control), or containing *Delftia tsuruhatensis*, and then infected with *L. major* parasites via artificial membrane feeding. At four days post-infection, groups of 50 sand flies (3 per condition, per experiment) were transferred to oviposition pots, and given sugar ad libitum for 10 days. After this period, sand fly carcasses were removed and the eggs allowed to hatch. Larvae were fed on a weekly basis until pupation. Then, all of the emerging adults were collected over a period of 2 weeks and counted. The bar graph represents the number of emerging adults sired by 150 sand fly females in the control (black bar) or bacteria-fed (red bar) groups. The group median and 95% CI values are plotted. Statistical significance

was determined using the Statistical significance was determined using the Mann-Whitney test and is highlighted: ns – not significant ($p > 0.05$). All results were obtained in three independent experiments.

*Q9: What are the effects of gut microbiota disruption caused by *Delftia tsuruhatensis* TC1 colonization on the growth and development of sandflies? What impact does it have on the growth of *Leishmania*? Why measure the metagenome of the midgut microbiome? What is the meaning?*

Response: Thank you for the questions. We decided to carry out a metagenomic analysis precisely because we were introducing a foreign bacterial agent in the midgut of our sand fly colony. Such an analysis would, on the one hand allow us to understand whether we could detect any changes in the abundance of *Delftia* spp. in the midgut of sand flies (and complement our microscopy results presented in Fig. 1), and on the other hand, to detect any changes in the sand fly gut microbiome composition. Do note that it is known that the sand fly gut microbiota influences the establishment of *Leishmania* within the sand fly midgut, since the sand flies treated with antibiotics (and consequently with an altered composition of their midgut microbiota) do not sustain *Leishmania* spp. infections [9, 11]. While this was briefly mentioned on the text of our original manuscript, we updated the sentence accordingly, as follows:

“Changes in the sand fly gut microbiota (e.g. after antibiotic treatment) can impact the establishment of *Leishmania* parasites in sand flies¹¹.”

*Q10: It may be considered to conduct field application experiments of *Delftia tsuruhatensis* TC1 in the wild or in parasitic-endemic areas to observe whether it can reduce the infection rate of *Leishmania*.*

Response: We completely agree with the Reviewer. This is actually our next step. We need to understand whether our results, obtained in the lab translate to the natural context. Initially, we plan to do experiments under controlled semi-field settings, in line with the approach we used in the context of mosquitoes[13]. These settings will allow us to investigate the initial potential of translatability to the field, which then needs to be confirmed in the context of actual field trials. Importantly, such settings will also allow us to optimize our deployment system to make sure the targeting of insects in nature will be successful. These notions were briefly discussed in our revised manuscript, as follows:

“...Considering that malaria and leishmaniasis are co-endemic in more than 50 countries worldwide^{42,43}, our data is of great relevance. Our results regarding sand flies, together with those recently reported with mosquitoes⁵, support the development of *Delftia tsuruhatensis* TC1-based interventions for the concurrent control of leishmaniasis and malaria in the field. Of note, while for mosquitoes, the targeting of the larval stage is an option, for sand flies such an approach would be quite challenging. Unlike mosquitoes, sand flies do not have an aquatic larval stage; they lay their eggs in organic matter-rich environments, which are difficult to locate and mostly unknown¹. Therefore, the reduced potential for bacterial dispersion in soil (versus water) makes deploying *Delftia tsuruhatensis* TC1 in such non-aquatic sites unfeasible. With this in mind, we envision, instead, the use of sugar baits adapted for targeted delivery, in places where the insects are expected to occur (e.g. breeding sites). Importantly, such sugar baits will not only contain the bacterial suspension for delivery, but also specific odorant-attracting molecules to make them more competitive than natural sugar sources (particularly in places where such sources are abundant), and

prevent potential sub-optimal deployment, as reported elsewhere in a different context⁴⁴. Of note, several molecules were reported to attract mosquitoes and sand flies, including some that have the potential to attract both insect species simultaneously (e.g. 1-octen-3-ol)^{45,46}. The development of these sugar baits is expected to be done in the near future, for use in the context of semi-field studies that will allow us the extrapolation of our results to the field context.”

*Q11: The study did not mention the principle and mechanism by which *Delftia tsuruhatensis* TC1 disrupts the transmission of *Leishmania*. From the author’s perspective, what could be the potential mechanisms and principles? Suggest adding and discussing the limitations of this study in the discussion.*

Response: Thank you for the question. Since we saw some effect of bacteria-excreted/secreted products on the growth of *Leishmania* parasites in vitro, and in parallel a major change in the gut microbiota of bacteria-fed sand flies (we remind the reviewer again that sand flies with a disrupted gut microbiota do not sustain good *Leishmania* infections), we think that our phenotype is due to a combination of factors: i) a direct effect of bacteria secreted/excreted products with leishmanicidal properties on *Leishmania* parasites, and ii) and an indirect effect of *D. tsuruhatensis*-induced sand fly gut dysbiosis on *Leishmania* parasites. This notion was emphasized in the discussion section of our revised manuscript.

Do note that the potential risks of our proposed intervention (limitations), which we briefly mentioned in the original version of our manuscript, were also emphasized in the revised version of the manuscript, as follows:

“Importantly, *Delftia* spp. are environmental bacteria, widely distributed in nature³³, and are part of the microbiota of different human-disease vectors, including sand flies, mosquitoes, and ticks^{11,34}. Therefore, the use of *D. tsuruhatensis* TC1 for the vector-based control of leishmaniasis is not expected to disturb natural ecosystems. However, the changes in the microbiota of wild hematophagous insects may be associated with risks, particularly when we consider the possibility of indirect pathogenic effects. We know that, together with *Leishmania* parasites, sand flies regurgitate a complex inoculum into the skin¹, including sand fly midgut bacteria³⁵. This has the potential to lead to *Leishmania* spp.-bacteria co-infection scenarios that are sporadically reported³⁶. Of note, most of these scenarios are probably just a consequence of the progression of leishmaniasis and not of vector bites-derived inoculation; visceral leishmaniasis is known to induce immune suppression³⁷ and cutaneous leishmaniasis may lead to the compromise of the skin barrier function³⁸, both risk factors for the establishment of opportunistic pathogens. Of relevance, the inoculation of gut bacteria via sand fly bites is dependent on the infection status; the higher the infection, the greater the probability of regurgitation to allow blood-feeding^{12,13}. Therefore, the fact that *D. tsuruhatensis* colonization leads to a decrease in the sand fly *Leishmania*-infection burden, may suggest these sand flies are egesting lower levels of gut bacteria into the host skin, making it less probable for these insects to transmit any resilient bacterial agent that becomes prevalent in a sand fly gut dysbiosis context. Of note, excluding CL lesions, no visible clinical manifestations were observed in our animals, including those exposed to the bites of *D. tsuruhatensis*-colonized sand flies. Adding to this, no *Delftia* secondary infection in the context of leishmaniasis was reported to date, whereas *Serratia* spp.-*Leishmania* spp. co-infection was reported only once³⁶. This said, in depth studies on the potential co-transmission of potentially pathogenic bacterial strains with *Leishmania* spp. parasites in the context *D. tsuruhatensis* TC1 colonization need to be carried out to support the development of any mitigation strategies, if needed.”

Reviewer #5

*This manuscript describes the exploration of *D. tsuruhatensis* infestation as a method of inhibiting the development of *Leishmania* species in sandflies. The authors note that this hasn't been explored before for sandflies but has been shown to be effective in mosquitoes. In this regard the results presented are novel and will be of significance in the field and have the potential for substantial global impact on public health. The paper is well laid out and easy to read. The methodology for the laboratory work seems to be complete and reproducible but this is not my area of expertise. The methodology for the modelling is less clear in the text and supplementary material and some clarification around the statistics is required but otherwise we believe the methodologies are sound and the results support the conclusions made in the paper. With the below (minor) adjustments the paper will be ready for publication.*

Response: We are grateful to Reviewer #5 for the time spent reading and evaluating this manuscript, and for the overall positive evaluation of our work. We are also thankful for the relevant comments that helped us to considerably improve our manuscript. Particularly, we thank the Reviewer the feedback on the clarity of our modelling studies; the feedback was quite useful. We considered all the comments and revised the manuscript (yellow highlighted text), when applicable; please check below our point-by-point responses to your comments.

1. Statistical significance: The manuscript text describes “significant” differences (sometime reductions/sometimes changes), but does not state clearly the p-value at which they consider significance. This seems to be 0.05. This should be stated in the text and consistently in figure captions (it currently differs figure to figure). The authors also need to be clear about what tests were used where. That is, Kruskal Wallis or Mann Whitney and what the alternative hypothesis were (i.e. two-tailed or one-tailed tests). This is technically different between a statistically significant ‘difference’ and a statistically significant increase/decrease. They also need to justify why they’ve used different tests.

Response: Thank you for the comment. Yes, our threshold for significance was always considered a p value equal or lower than 0.05. Do note that this was disclosed in the original Materials and Methods section of our manuscript (“The nonparametric Mann-Whitney or Kruskal-Wallis tests, the latter with post-hoc analysis, were used to access statistical differences, with at least $P \leq .05$.”). For the sake of clarity, we updated this sentence as follows:

“The nonparametric Mann-Whitney or Kruskal-Wallis tests, the latter with post-hoc analysis (Dunn’s test), were used to access statistical differences. A p value ≤ 0.05 was considered statistically significant.”

This said, now, we clearly mention the p values in the manuscript text and uniformized the figure captions (having in mind length limitations). As per the question on our mention of significant differences *versus* increases/decreases, indeed, now that we have double-checked, some are not accurate. We confirmed our statistical analyses thoroughly and updated the text for accuracy. Here is the tests used to check for statistical relevance in our data-sets:

- Revised Fig. 2: Mann-Whitney test (Two-tailed) – manuscript text changed accordingly [‘e.g. instead of “Bacteria-fed sand flies showed a significant reduction in the total number of parasites ...” now we write “Bacteria-fed sand flies showed a reduction in the total number of parasites per midgut, 7-, and 11-days post-infection with statistical significance (*versus* controls; $p \leq 0.0005$; Fig. 2b)”].]
- Revised Fig. 3a-c: Mann-Whitney test (Two-tailed) – no need to update the manuscript text.

- Revised Fig. 3d: Kruskal-Wallis with Dunn's post-hoc analysis – no need to update the manuscript text.
- Revised Fig. 4: Mann-Whitney test (Two-tailed) – manuscript text changed accordingly [‘e.g. instead of “However, 4 days later (11 days post-infection / 6 days bacterial feeding), a significant reduction in the number of parasites...” now we write “However, 4 days later (11 days post-infection / 6 days bacterial feeding), a reduction in the number of parasites per midgut in bacteria-fed *versus* control sand flies was observed, with statistical significance (>90% reduction; $p \leq 0.0005$; Fig. 4b, right graph).”.]
- Revised Fig. 5a: Mann-Whitney test (Two-tailed) – no need to update the manuscript text.
- Revised Fig. 5d: Kruskal-Wallis with Dunn's post-hoc analysis – no need to update the manuscript text.
- Revised Fig. 6 a and d: Mann-Whitney test (Two-tailed) – manuscript text changed accordingly [‘e.g. instead of “Expectedly, the percentage of blood-engorged sand flies in the bacteria-fed group was significantly higher...” now we write “Expectedly, the percentage of blood-engorged sand flies in the bacteria-fed group was higher than that in the control group, with statistical significance ($p=0.0016$; Fig. 6a).”.]

2. *Mathematical model: The mathematical model isn't specified at all in the main text of the manuscript, nor is the appropriate reference to the model source supplied. This is supplied in the supplemental material, but although the R code runs it is difficult to see where the values used actually come from (they are not calculated but are hard coded to produce the figures). The model and the results need to be clarified so that the model based conclusions are properly supported.*

Response: Thank you for the comment. After re-examining the manuscript, we agree that there is room for improved with regards to our modelling studies. In the revised version of our manuscript (Methods), now we include a more detailed explanation of the calculation of the R_0 values (including the equations used and the relevant References that support them). Additionally, we also clarify the assumptions we made to estimate the potential impact of the introduction of our bacterial strain in the context of a *Leishmania* transmission ecosystem (also in the revised Methods section). Furthermore, we updated our code, for the sake of transparency. We provide more information to explain in detail each step (including any relevant assumptions made) and include the raw data that we used to estimate the changes in R_0 values induced by alteration in vector mortality and *Leishmania* sand fly-transmission in the context of *D. tsuruhatensis* colonization.

Specific comments:

P4L69: “significant reduction” The authors later state that this is a Mann-Whitney test. Is the test two sided? If so it is a significant change. The test itself hasn't tested for reduction/increase. At what level of significance?

Response: Thank you for the question. Indeed, this test was carried out as a 2-tailed test, and thus, the Reviewer is correct. We apologize for this oversight. The sentence was corrected in accordance, for accuracy, as follows:

“Bacteria-fed sand flies showed a reduction in the total number of parasites per midgut, 7-, and 11-days post-infection with statistical significance (*versus* controls; $p \leq 0.0005$; Fig. 2b).”

P8L173: Is it worth discussing what effect increased mortality has on control programs? Will this mean that the control needs to be re-introduced (i.e. infected sandflies die off so the infection isn't self-sustaining).

Response: Thank you for the question. We would like to start by noting that the increased mortality was only observed (with statistical significance) in the context of *Leishmania*-infected sand flies. It is important to mention that the proportion of infected flies in the field is low (ranging from 0.7% to 2% according to different studies, although this may be an underestimation [12]). Therefore, the decreased fitness of infected sand flies is not expected to lead to dramatic changes in the overall population. This, however, needs to be confirmed in the context of semi-field studies, which is our next step. Regarding the "effect on control programs" and the question of whether this bacterial strain needs to be re-introduced, we believe the answer is yes. Our published data on laboratory-reared mosquitoes suggest that this bacterial species is not transmitted vertically [13]. We have some preliminary data that may suggest the same is true for sand flies, which again needs to be confirmed in the context of semi-field studies involving wild-caught sand flies. This is also part of our next steps. Importantly, while the need for re-introduction of a control agent may be perceived as a "weakness," we think it can be a "strength." *Delftia tsuruhatensis* TC1 can be used in both "highly endemic" and "outbreak" contexts until transmission cycles are stopped, and then be discontinued with no residual impact (and pressure) on the ecosystem. It is worth noting that organisms tend to adapt to external continuous pressures (e.g., thermotolerance of typically non-pathogenic organisms due to global warming [14]).

P10L206: where does the model of R_0 come from? Is it realistic? What are the assumptions/limitations. This 'model' needs to be better described here.

Response: Thank you for the questions. The model is based on a multi-host setting. For the sake of clarity, we updated the Methods section of our revised manuscript. Now we refer to the modelling literature where the R_0 values were derived from, as well as to the important equations we used for the analysis. Furthermore, now we also explicitly mention the different assumptions we made, for instance the notion that we assumed the results obtained in the laboratory would be translatable to the field context. Please check our revised Methods section (Modelling sub-section) for further details.

P11L224: "presumably low ecological impact". This links to comment P8L173. If you are changing their mortality rate can that have a flow on impact?

Response: Thank you for the question. First of all, we used the conditional, precisely because we do not know exactly the real impact of such an intervention on sand fly populations in the field. As mentioned above, we are planning some semi-field studies to answer to this and other relevant questions in detail. This said, our idea was to use the expression "*presumably low ecological impact*" in a comparative context, which we now see was not successful. Comparing to the sand fly control measures in effect today, this approach, that only kills infected sand flies (and not all the sand flies), has a lower ecological impact. We revised the Discussion section of our manuscript for clarity, as follows:

"While human interventions should minimize the disruption of ecosystems¹⁸, the targeting of specific disease-transmitting arthropod populations is admissible from a risk-benefit perspective. In fact, many

vector control approaches are specifically designed to impact the fitness of disease insect vectors^{4,19}. This is particularly true in the context of sand flies. Sand fly-based interventions for the control of leishmaniases rely mostly (if not only) on the use of insecticides, whose efficacy is limited not only because of the emergence of resistance, but also because we still lack basic epidemiologic information²⁰. To the best of our knowledge, there are still no strategies that promote sand fly-vector refractoriness for the control of leishmaniasis. Here, we show that *D. tsuruhatensis* TC1 can colonize the gut of sand flies, leading to significant changes that negatively impact the transmission of *Leishmania* to hosts. Importantly, we also show that this bacterial strain seems to impact the fitness of *Leishmania*-infected sand flies alone (accounting to a total of 0.7 to 2% of all sand flies in the field, according to different studies²¹), as opposed to the indiscriminate killing of all sand flies, regardless of the infection status by insecticides. Therefore, the use of *D. tsuruhatensis* TC1 is a promising vector-based approach for the control of leishmaniasis in the field, with a presumably lower ecological impact than the vector control approaches in effect today.”

Figure 1 C and D (and others): The confidence intervals are hidden by the points. It may be worthwhile to change the confidence intervals to a shade of grey and place them on top of the points for visibility. An alternative may be violin plots which would give the same effect as the jittering of points.

Response: Thank you for the comment. Accordingly, we revised all relevant Figures and make sure that now, in the revised files, all confidence intervals are visible.

*P20L500 (and other figure captions): It would be a good idea to have the ns >0.05; * <0.05 included in the captions and specified in the text for consistency.*

Response: Thank you for the comment. All relevant captions were updated to include this detail, as suggested by the Reviewer.

Figure 2C and Figure 4D and captions: It's not clear what categories are being tested and which test is being conducted.

Response: Thank you for the question. For both analyses we used the Kruskal-Wallis test followed by a multiple comparison post-hoc analysis that corrects for type I errors (Dunn's test). In the first case, every group was compared with every group because such comparisons were relevant. In the second case, the relevant comparisons were pre-chosen: Control No Blood *versus* Control Non-infected Blood, Control Infected Blood, and Bacteria No Blood; Control Non-infected Blood *versus* Control Infected Blood, and Bacteria Non-infected Blood; Control Infected Blood *versus* Bacteria Infected Blood; Bacteria No Blood *versus* Bacteria Non-infected Blood, and Bacteria Infected Blood; and Bacteria Non-infected Blood *versus* Bacteria Infected Blood. Do note that the highlighted differences are those given by the post-hoc analysis, and not those given by the actual Kruskal-Wallis test; of note, in the second graph we only highlight statistically significant differences for the sake of clarity. We made sure that this is now clearer in the respective Figure legends.

Remarks on code availability

Most of the code is made available at a github repo. One of us has looked over the code, but have no experience with the specific software used apart from R scripts. Data files are not part of the git repo, so code could not be effectively re-run as far as I could tell. R code for the model of R0 is provided in the supplementary and runs to reproduce the R0 figure (figure 53E). However, values for the plots are hard coded and it isn't always clear where the values come from.

Response: We would like to thank the Reviewer for the thorough examination our codes to ensure reproducibility. As per the code referring to the metagenomic analysis, unfortunately, we cannot include any data files as a part of the GitHub repository, as they must remain confidential prior to publication. To comply with Nature journals' requirements for data availability, we have deposited all raw data in NCBI under BioProject PRJNA1079352; data release is pending to after the manuscript is published. This said, to provide the Reviewer complete access to our raw data, we now provide a reviewer-access link (<https://dataview.ncbi.nlm.nih.gov/object/PRJNA1079352?reviewer=6i7uegkn847himbdis15bg511m>). The results can be fully reproduced using the raw data deposited and the analysis scripts made available in our GitHub repository, as described in detail in the Methods and Supplemental Information.

Additionally, with respect to our modelling studies, we agree that it could be improved for the sake of transparency. In the revised R code, now we include the raw data used to estimate changes in vector mortality and parasite transmission, the 2 parameters used to estimate the potential impacts of the bacteria treatment in the endemicity of leishmaniasis. Do note that we updated the code so that parameters are estimated within the code, using the data from our experiments; then we use such estimates to obtain the figures where R0 changes are depicted.

Reviewer #6

Response: We would like to thank Reviewer #6 for the time spent reading and evaluating this manuscript. Peer-review is an essential step in the scientific process and the feedback from our peers is always very important. We are sure your evaluation raised some important points that in the end allowed us to re-submit a clearer manuscript, and for that, we are thankful. We revised the manuscript in line with the issues raised by all Reviewers (yellow highlighted text), when applicable. Please check our responses to the issues raised by all Reviewers', above, and check if we addressed your concerns adequately.

References

1. Han J, Sun L, Dong X, Cai Z, Sun X, Yang H, et al. Characterization of a novel plant growth-promoting bacteria strain *Delftia tsuruhatensis* HR4 both as a diazotroph and a potential biocontrol agent against various plant pathogens. *Syst Appl Microbiol*. 2005;28(1):66-76. doi: 10.1016/j.syapm.2004.09.003. PubMed PMID: 15709367.
2. Malesevic M, Di Lorenzo F, Filipic B, Stanisavljevic N, Novovic K, Senerovic L, et al. *Pseudomonas aeruginosa* quorum sensing inhibition by clinical isolate *Delftia tsuruhatensis* 11304: involvement of N-octadecanoylhomoserine lactones. *Sci Rep*. 2019;9(1):16465. Epub 20191111. doi: 10.1038/s41598-019-52955-3. PubMed PMID: 31712724; PubMed Central PMCID: PMC6848482.
3. Yin Z, Liu X, Qian C, Sun L, Pang S, Liu J, et al. Pan-Genome Analysis of *Delftia tsuruhatensis* Reveals Important Traits Concerning the Genetic Diversity, Pathogenicity, and Biotechnological Properties of the Species. *Microbiol Spectr*. 2022;10(2):e0207221. Epub 20220301. doi: 10.1128/spectrum.02072-21. PubMed PMID: 35230132; PubMed Central PMCID: PMC9045143.
4. Cecilio P, Cordeiro-da-Silva A, Oliveira F. Sand flies: Basic information on the vectors of leishmaniasis and their interactions with *Leishmania* parasites. *Commun Biol*. 2022;5(1):305. Epub 20220404. doi: 10.1038/s42003-022-03240-z. PubMed PMID: 35379881; PubMed Central PMCID: PMC979968.
5. Potter CJ. Stop the biting: targeting a mosquito's sense of smell. *Cell*. 2014;156(5):878-81. Epub 2014/03/04. doi: 10.1016/j.cell.2014.02.003. PubMed PMID: 24581489.
6. Tavares DDS, Salgado VR, Miranda JC, Mesquita PRR, Rodrigues FM, Barral-Netto M, et al. Attraction of phlebotomine sandflies to volatiles from skin odors of individuals residing in an endemic area of tegumentary leishmaniasis. *PLoS One*. 2018;13(9):e0203989. Epub 2018/09/25. doi: 10.1371/journal.pone.0203989. PubMed PMID: 30248113; PubMed Central PMCID: PMC6152958.
7. Wang S, Jacobs-Lorena M. Genetic approaches to interfere with malaria transmission by vector mosquitoes. *Trends Biotechnol*. 2013;31(3):185-93. Epub 20130206. doi: 10.1016/j.tibtech.2013.01.001. PubMed PMID: 23395485; PubMed Central PMCID: PMC3593784.
8. Fraihi W, Fares W, Perrin P, Dorkeld F, Sereno D, Barhoumi W, et al. An integrated overview of the midgut bacterial flora composition of *Phlebotomus perniciosus*, a vector of zoonotic visceral leishmaniasis in the Western Mediterranean Basin. *PLoS Negl Trop Dis*. 2017;11(3):e0005484. Epub 20170329. doi: 10.1371/journal.pntd.0005484. PubMed PMID: 28355207; PubMed Central PMCID: PMC5386300.
9. Louradour I, Monteiro CC, Inbar E, Ghosh K, Merkhofer R, Lawyer P, et al. The midgut microbiota plays an essential role in sand fly vector competence for *Leishmania major*. *Cell Microbiol*. 2017;19(10). Epub 20170619. doi: 10.1111/cmi.12755. PubMed PMID: 28580630; PubMed Central PMCID: PMC5587349.
10. Tabbabi A, Mizushima D, Yamamoto DS, Kato H. Sand Flies and Their Microbiota. *Parasitologia*. 2022;2:71-87. doi: <https://doi.org/10.3390/parasitologia2020008>.
11. Kelly PH, Bahr SM, Serafim TD, Ajami NJ, Petrosino JF, Meneses C, et al. The Gut Microbiome of the Vector *Lutzomyia longipalpis* Is Essential for Survival of *Leishmania infantum*. *mBio*. 2017;8(1). Epub 20170117. doi: 10.1128/mBio.01121-16. PubMed PMID: 28096483; PubMed Central PMCID: PMC5241394.
12. Tiwary P, Kumar D, Mishra M, Singh RP, Rai M, Sundar S. Seasonal variation in the prevalence of sand flies infected with *Leishmania donovani*. *PLoS One*. 2013;8(4):e61370. Epub 2013/04/16. doi: 10.1371/journal.pone.0061370. PubMed PMID: 23585896; PubMed Central PMCID: PMC3621828.
13. Huang W, Rodrigues J, Bilgo E, Tormo JR, Challenger J, de-Cozar-Gallardo C, et al. *Delftia tsuruhatensis* TC1 symbiont suppresses malaria transmission by anopheline mosquitoes. *Science*. 2023;381(6657):533-40. doi: 10.1126/science.adf8141.

14. Seidel D, Wurster S, Jenks JD, Sati H, Gangneux JP, Egger M, et al. Impact of climate change and natural disasters on fungal infections. *Lancet Microbe*. 2024;5(6):e594-e605. Epub 20240319. doi: 10.1016/S2666-5247(24)00039-9. PubMed PMID: 38518791.

Rebuttal Letter - R2

Reviewer #1

I appreciate the effort authors made on the manuscript; they considered all the comments and revised carefully the text. I recommend the manuscript for publication in Nature Communications.

Response: Once again, we would like to thank Reviewer #1 for the time spent reading and evaluating our manuscript. We are glad we were able to address all your concerns properly and are happy you now find our manuscript suitable for publication in Nature Communications. We are convicted that the relevant issues raised in the previous revision round helped us to considerably improve our manuscript. For that, we thank you once more.

Reviewer #2

After carefully reviewing the revised version of the manuscript by Cecilio et al., I note that all my comments and suggestions have been addressed. The study appears to be scientifically well-designed, and the data generated is well-analyzed and highly interesting. The discussion surrounding the data is also well-articulated.

Response: Again, we would like to thank Reviewer #2 for the time spent reading and evaluating our manuscript. We are glad we were able to address all your concerns properly. We do also feel the paper reads well. Do note that we are convicted that by answering to the relevant issues raised by you in the previous revision round, our manuscript has improved. For that, we thank you one last time.

Reviewer #4

After reviewing the revised manuscript submitted by the author, I believe that the author has carefully addressed the issues and suggestions I raised. I believe that the author's revised manuscript has met the standards for publication. Therefore, I agree and recommend accepting the publication of this article.

Response: Once again, we would like to thank Reviewer #4 for the time spent reading and evaluating our manuscript. We are glad we were able to address all your concerns properly and are happy you now find our manuscript suitable for publication in Nature Communications. We are convicted that the relevant issues raised in the previous revision round helped us to considerably improve our manuscript. For that, we thank you once more.

Reviewer #7

As requested by the editor, I have reviewed the statistical and modelling methods of this paper. The authors have addressed most of the corrections suggested by Reviewer #5. However, I recommend further revisions to ensure the manuscript is suitable for publication:

Response: First, we want to thank Reviewer #7 for the time spent reading and evaluating this manuscript. The authors appreciate the thorough revision of the statistical and modelling methods on top of a previous revision by Reviewer #5. We are glad the Reviewer #7 thinks most concerns previously raised were addressed. We are also thankful for these relevant comments that helped us to considerably improve our manuscript. We considered all of the issues raised by the Reviewer and revised the manuscript (yellow highlighted text), when applicable; please check below our point-by-point responses to your specific comments.

- Apply one-tailed versions of the non-parametric tests or correct the current interpretation of two-tailed tests.

Response: Thank you for this comment. We decided to correct the interpretation in lieu of apply one-tailed versions of the statistical tests. Please see our answers below for a detailed explanation.

- Enhance the legend of Figure 6e for better clarity.

Response: Thank you for this suggestion. We updated the respective legend, accordingly. Please read our answers below for more details.

- Correct a minor typo in the R code.

Response: Thank you for highlighting this typo; we apologize for this oversight. The R code was corrected accordingly. Please read our answers below for more details.

Please read below for more details:

1) Statistical significance tests

Reviewer #5 raised a valid point regarding the need for additional details on the Mann-Whitney and Kruskal-Wallis tests. The authors have addressed this concern in the current version of the manuscript, making it clearer where non-parametric tests were applied. However, the Methods section could still be enhanced by providing a rationale for selecting these tests over others, such as the t-test, particularly in the context of the nature of the data (parametric versus non-parametric).

Response: Thank you for this comment. Our initial analysis plan was based on non-parametric tests because our datasets were designed to be of less than 30 samples per group. As suggested by the reviewer we now tested the normality of all datasets (Source data). In a few datasets all groups showed normal distributions. Thus, we revised the statistical analyses referring to these (Figure 2e, Figure 3a-c, Figure 6a,

Figure S2b, and Figure S2c) and use the parametric Unpaired t-test instead; all relevant Figure Legends were updated accordingly. Do note this re-analysis did not lead to any changes in the interpretation of our results (significant results remained significant and non-significant results remained non-significant). In line with this, following your suggestion, we also updated the text in the “Data representation and statistics” subsection in our Methods section, for clarity, as follows:

“Results obtained in at least 2 independent experiments are shown per individual sand fly/mouse, with a representation of the group mean/median value \pm 95% confidence interval (CI), unless otherwise stated. Statistical analysis was performed using GraphPad Prism software v6.01. All datasets were first subjected to normality tests (Shapiro-Wilk or Kolmogorov-Smirnov). The Unpaired t-test (parametric; always two-tailed) was used to access statistical differences when all groups in a dataset showed normal distributions. The Mann-Whitney test (nonparametric; always two-tailed) or Kruskal-Wallis test, the latter with post-hoc analysis (Dunn’s test), were used to access statistical differences when at least one group in a dataset did not show a normal distribution. A p value \leq 0.05 was considered statistically significant.”

There are inconsistencies in the application of these tests. The distinction between two-tailed and one-tailed tests is not clearly addressed, which may lead to potential misinterpretations of results. Two-tailed tests identify any difference between groups regardless of direction (increase or decrease), while one-tailed tests specifically determine if a treatment is better or worse.

Response: Thank you for the comment. We always used **two-tailed tests** in our analyses. Do note, that these details were all provided in the source file submitted together with our paper. In case the reviewer does not have access to the source file, here is a list of statistical tests used for the analysis of the relevant data in our main figures:

- Revised Fig. 2: panels **b-d**, Mann-Whitney test (**two-tailed**); panel **e** (data in both groups following a normal distribution), Unpaired t-test (**two-tailed**).
- Revised Fig. 3: panels **a-c** (data in both groups following a normal distribution), Unpaired t-test (**two-tailed**); panel **d**: Kruskal-Wallis with Dunn’s post-hoc analysis (**two-tailed** nature can be attributed to the Dunn’s test).
- Revised Fig. 4: **all panels**, Mann-Whitney test (**two-tailed**)
- Revised Fig. 5: panel **a**, Mann-Whitney test (**two-tailed**); panel **d**, Kruskal-Wallis with Dunn’s post-hoc analysis (**two-tailed** nature can be attributed to the Dunn’s test).
- Revised Fig. 6: panel **a** (data in both groups following a normal distribution), Unpaired t-test (**two-tailed**); panel **d**, Mann-Whitney test (**two-tailed**).

For clarity, now we mention in our methods section that all tests used in the analysis of our data were two-tailed (when applicable):

“The Unpaired t-test (parametric; always two-tailed) was used to access statistical differences when all groups in a dataset showed normal distributions. The Mann-Whitney test (nonparametric; always two-tailed) or Kruskal-Wallis test, the latter with post-hoc analysis (Dunn’s test), were used to access statistical differences when at least one group in a dataset did not show a normal distribution.”

On the current version of the manuscript, with the application of the two-tailed tests, the authors should correct all sentences mentioning statistical significance, as the following example:

“Bacteria-fed sand flies showed a reduction in the total number of parasites per midgut, 7-, and 11-days post-infection with statistical significance (versus controls; $p \leq 0.0005$; Fig. 2b)” should be corrected to: “Bacteria-fed sand flies showed significant difference in the total number of parasites per midgut, 7-, and 11-days post-infection (versus controls; $p \leq 0.0005$; Fig. 2b)”

Response: Thank you for the suggestion. We edited the manuscript text as below for accuracy:

- Revised Fig. **2b** – “Bacteria-fed sand flies showed a reduction in the total number of parasites per midgut, 7-, and 11-days post-infection; **of note, this difference was statistically significant** (versus controls; $p \leq 0.0005$; Fig. 2b), and a dose-dependence tendency was observed in this context (fig. S1).” *Do note that we are saying the difference (and not the reduction) was significant.*
- Revised Fig. **2c-d** – no changes needed.
- Revised Fig. **2e** – “Interestingly, when we compared the diameter of the anterior midgut of *D. tsuruhatensis*-fed versus control sand flies, we observed that the former showed smaller diameters; **such difference was statistically significant** ($p=0.0266$; Fig. 2e).”
- Revised Fig. **3** – no changes needed.
- Revised Fig. **4b** – “However, 4 days later (11 days post-infection / 6 days bacterial feeding), a reduction in the number of parasites per midgut in bacteria-fed versus control sand flies was observed; **this difference was statistically significant** (>90% reduction; $p < 0.0001$; Fig. 4b, right graph).”
- Revised Fig. **4a, c, d** – no changes needed.
- Revised Fig. **5** – no changes needed.
- Revised Fig. **6a** – “Expectedly, the percentage of blood-engorged sand flies in the bacteria-fed group was higher than that in the control group; **of note, this difference was statistically significant** ($p=0.0005$; Fig. 6a).”
- Revised Fig. **6b-e** – no changes needed.
- Revised Fig. **S1** – no changes needed.
- Revised Fig. **S2b**: “Parasites **showed impaired growth**, when cultured with at least 50% *D. tsuruhatensis*-supernatant (versus control supernatant; at least $p \leq 0.05$; fig. S2a, b).”
- Revised Fig. **S2a, c-g**: no changes needed.
- Revised Fig. **S3** – no changes needed.
- Revised Fig. **S4** – no changes needed.

However, I highly recommend replacing these tests by their one-tailed versions, so that clear conclusions about the direction of the experiments (increase/decrease) can be drawn.

Response: Thank you for this suggestion. We could opt to re-analyze the data using one-tailed tests, but we think not doing it is more correct. As the Reviewer knows, statistical analysis must be planned a priori, a rule that we followed. Because we did not know what to expect (either a decrease or increase in the number of parasites per midgut in response to bacterial colonization, both previously reported^{1,2}), we decided to use one test that was not directional. In the future, now that we know what direction to expect, we will definitely consider the use of one-tailed statistical tests in a similar context.

As additionally requested by Reviewer #5, the confidence intervals are now clearly visible in the plots.

Response: Thank you for the comment. We are glad these changes were satisfactory.

2) Mathematical model (R_0)

The new section in the Methods about the R_0 models is a clear improvement as suggested by Reviewer #5, as details about this approach were missing on a previous version of the manuscript. The text now adequately describes how the model was applied, its assumptions and limitations. Upon reviewing this section, I understood the model and I agree that its results support the conclusions about the impact of the bacterium treatment on leishmaniasis transmission.

Response: Thank you for the positive evaluation. We are glad the Reviewer thinks our changes met the expectations.

The legend of Figure 6e can be improved for clarity. I would suggest clearly mentioning the difference between the top and bottom panels (earlier in the legend text), and that the diagonal lines and colour shades represent the R_0 values.

Response: Thank you for this suggestion. We agree with the reviewer that the Figure Legend could be even clearer. Do note that we have a limited number of words for legends, we already surpassed. This said, we included the mentioned details in our revised Figure Legend, as follows:

“(e) Basic Reproduction Number (R_0) surfaces were generated as function of the infection rate (β^2) and 14-day mortality (μ) for a leishmaniasis transmission model with reservoirs/incidental hosts^{47,48}, as per the equation $R_0 = C \frac{\beta^2}{\mu}$ where C is a parameter that is a function of reservoir host and vector relative abundance, which we set at $C = 2$ in the plot. On the surface, black circles represent R_0 estimates from three datasets, a cross sectional serosurvey in dogs¹⁴ (Dogs, $\widehat{R}_0 = 1.20$), an outbreak investigation with multiple vertebrate hosts/vectors¹⁶ (Multiple Hosts, $\widehat{R}_0 = 1.27$), and the analysis of a time-series of human cases¹⁷ (Humans, $\widehat{R}_0 = 1.90$). Diagonal lines represent parameter combinations with the same R_0 value, while different shades of orange represent ranges between the values demarked by the diagonal lines. To locate R_0 estimates on the surface we assumed infection and mortality as determined for our experimental controls ($\widehat{\beta}^2 = 0.51$ and $\widehat{\mu} = 0.28$). White arrows represent the reduction in R_0 when parameters changed based on the treatment results from our experiments, with mortality increasing alone ($\widehat{\mu} = 0.44$; top panel), or together with reduced transmission potential (bottom panel; $\widehat{\beta}^2 = 0.33$). R_0

estimates were reduced as follows: Dogs ($\widehat{R}_0 = 0.75$, or $\widehat{R}_0 = 0.45$), Multiple Hosts ($\widehat{R}_0 = 0.80$, or $\widehat{R}_0 = 0.48$) and Humans ($\widehat{R}_0 = 1.19$, or $\widehat{R}_0 = 0.72$).”

An additional suggestion, though not mandatory, is to incorporate new points at the ends of the arrows. This would clearly illustrate to the reader the updated values of the mortality and transmission rates associated with the R_0 values in the control scenarios. If the authors would like to add the points to Fig 6e, I suggest adding the following in the end of each plot code:

```
# top panel
points(c((mu_x_b_14),(mu_x_b_14),(mu_x_b_14))~c(1.20*(mu_x_c_14),1.271*(mu_x_c_14),1.90*(mu_x_c_14)),pch=19,cex=0.75)
# bottom panel
points(c((mu_x_b_14),(mu_x_b_14),(mu_x_b_14))~c(1.20*(mu_x_c_14)*beta_change,1.271*(mu_x_c_14)*beta_change,1.90*(mu_x_c_14)*beta_change),pch=19,cex=0.75)
```

And then slightly move the arrow tips so they don't overlap with the new points:

```
# top panel
arrows(x0=1.20*(mu_x_c_14),y0=(mu_x_c_14),x1=1.20*(mu_x_c_14),y1=(mu_x_b_14-0.01),lwd=2,col="white",length = 0.075)
arrows(x0=1.271*(mu_x_c_14),y0=(mu_x_c_14),x1=1.271*(mu_x_c_14),y1=(mu_x_b_14-0.01),lwd=2,col="white",length = 0.075)
arrows(x0=1.90*(mu_x_c_14),y0=(mu_x_c_14),x1=1.90*(mu_x_c_14),y1=(mu_x_b_14-0.01),lwd=2,col="white",length = 0.075)

# bottom panel
arrows(x0=1.20*(mu_x_c_14),y0=(mu_x_c_14),x1=1.20*(mu_x_c_14)*beta_change+0.002,y1=(mu_x_b_14),lwd=2,col="white",length = 0.075)
arrows(x0=1.271*(mu_x_c_14),y0=(mu_x_c_14),x1=1.271*(mu_x_c_14)*beta_change+0.002,y1=(mu_x_b_14),lwd=2,col="white",length = 0.075)
```

```
arrows(x0=1.90*(mu_x_c_14),y0=(mu_x_c_14),x1=1.90*(mu_x_c_14)*beta_change+0.002,y1=(mu_x_b_14),lwd=2,col="white",length = 0.075)
```

Response: Thank you for the suggestion. We ran this code, and got the figures, as you can see below:

Looking at both figures side-by-side, we do prefer our previous version, reason why we kept it. It is purely a stylistic preference.

3) Code reproducibility

I was also able to fully reproduce the provided R code for the application of R0 models and the respective plot design (Fig 6e), except for a minor typo at line 80 of the code (capital X on the l_x_bacteria object):

```
#p_x_bacteria= l_X_bacteria[-1]/l_x_bacteria[-length(l_x_bacteria)]  
p_x_bacteria= l_x_bacteria[-1]/l_x_bacteria[-length(l_x_bacteria)]
```

The R code is well commented, and the sources of the hard-coded values are clearly explained. I suggest uploading it to a code repository (GitHub, GitLab) to facilitate reproducibility and skipping the step of manually copy/pasting it from the supplementary document file.

Response: Thank you for the comments and positive evaluation. Thank you also for finding the typo. We apologize for this oversight. We corrected it accordingly, as follows:

...

```
l_x_bacteria = c(100,  
                97.06322581,  
                94.65354839,  
                88.49129032,  
                84.74967742,  
                82.66612903,  
                81.29032258,  
                79.47354839,  
                79.19129032,  
                74.83290323,  
                72.94290323  
)
```

```
p_x_bacteria=l_x_bacteria[-1]/l_x_bacteria[-length(l_x_bacteria)]  
mu_x_bacteria=mean(-log(p_x_bacteria))
```

...

As per depositing this code in GitHub, we decided not to because it is so simple. We do have a GitHub page with the analysis pipeline used for the analysis of the metagenomics data.

Like Reviewer #5, I lack the expertise in bioinformatics to assess the reproducibility of the metagenomic analysis code on the GitHub repository.

Response: Thank you for this comment. Please rest assured that all metagenomics analyses were performed by a trustworthy collaborator with extensive expertise on the field.

References

- 1 - Louradour, I. *et al.* The midgut microbiota plays an essential role in sand fly vector competence for *Leishmania major*. *Cell Microbiol* **19** (2017). <https://doi.org/10.1111/cmi.12755>
- 2 - Campolina, T. B., Villegas, L. E. M., Monteiro, C. C., Pimenta, P. F. P. & Secundino, N. F. C. Tripartite interactions: *Leishmania*, microbiota and *Lutzomyia longipalpis*. *PLoS Negl Trop Dis* **14**, e0008666 (2020). <https://doi.org/10.1371/journal.pntd.0008666>